# SALM1 controls synapse development by promoting F-actin/PIP2-dependent Neurexin clustering

Marinka Brouwer[1], Fatima Farzana[2], Frank Koopmans[2,3] , Ning Chen[2,3], Jessie W Brunner[2], Silvia Oldani[2], Ka Wan Li[3], Jan RT van Weering[1], August B Smit[3], Ruud F Toonen[2,*] & Matthijs Verhage[1,2,**]

## Abstract

Synapse development requires spatiotemporally regulated recruitment of synaptic proteins. In this study, we describe a novel presynaptic mechanism of *cis*-regulated oligomerization of adhesion molecules that controls synaptogenesis. We identified synaptic adhesion-like molecule 1 (SALM1) as a constituent of the proposed presynaptic Munc18/CASK/Mint1/Lin7b organizer complex. SALM1 preferentially localized to presynaptic compartments of excitatory hippocampal neurons. SALM1 depletion in excitatory hippocampal primary neurons impaired Neurexin1β- and Neuroligin1-mediated excitatory synaptogenesis and reduced synaptic vesicle clustering, synaptic transmission, and synaptic vesicle release. SALM1 promoted Neurexin1β clustering in an F-actin- and PIP2-dependent manner. Two basic residues in SALM1's juxtamembrane polybasic domain are essential for this clustering. Together, these data show that SALM1 is a presynaptic organizer of synapse development by promoting F-actin/PIP2-dependent clustering of Neurexin.

**Keywords** neurexin; PIP2; SALM1; synapse organization; synaptogenesis
**Subject Category** Neuroscience
**The EMBO Journal (2019) 38: e101289**

## Introduction

Establishing brain connectivity involves the precise targeting of billions of axons to their specific targets followed by the development of functional synapses. A large collection of cell adhesion molecules (CAMs) help orchestrate this complex connectivity and the assembly of synapses between specific populations of neurons (Robbins *et al*, 2010; Yim *et al*, 2013; Chen *et al*, 2017; Jiang *et al*, 2017). The selective expression of different CAMs is considered a major determinant of synaptic diversity (Fuccillo *et al*, 2015; Foldy *et al*, 2016; de Wit & Ghosh, 2016). However, how these individual components work together to establish brain connectivity is still largely unknown.

CAMs form homo- and/or heteromeric *trans*-cellular interactions between their extracellular domains that are often sufficient to induce synapses (Scheiffele *et al*, 2000; Biederer *et al*, 2002; Graf *et al*, 2004; Yim *et al*, 2013). In addition, CAMs can form homo- and heteromeric *cis*-interactions via their extracellular domains that result in CAM oligomerization. *Cis*- and *trans*-interactions can be competitive or cooperative resulting in a complex interplay between different CAMs and their downstream effectors in individual synapses (Aricescu & Jones, 2007; Taniguchi *et al*, 2007; Lie *et al*, 2016). Multiple different CAMs are thought to *cis*-localize to individual synapses and cooperate to control synaptogenesis. Inhibition of direct homomeric *cis*-interactions and oligomerization of specific CAMs (Neuroligin, SynCAM, and SALM5) impaired *trans*-interactions and synapse development (Fogel *et al*, 2011; Shipman & Nicoll, 2012; Lin *et al*, 2018), suggesting that *cis*-interactions and CAM oligomerization control *trans*-interactions and synapse development. Many CAMs bind intracellular scaffolding proteins including PSD95, SAP102, CASK, and liprin-α which recruit and stabilize other synaptic proteins and are proposed to link cell adhesion to synapse assembly (Dalva *et al*, 2007). How different CAMs are organized in individual synapses and how they organize pre- and postsynaptic specializations is still poorly understood.

In this study, we aimed to elucidate how CAMs organize *trans*-synaptic signaling, link to intracellular scaffolding proteins, and regulate synapse formation using the presynaptic organizer complex CASK/Mint1/Lin7b as a starting point. We identified the cell adhesion molecule SALM1 as an interactor of this complex. Other members of the SALM family, SALM2-5, have previously been described as postsynaptic proteins (Lie *et al*, 2018) and regulate synapse formation by postsynaptic mechanisms, but the function of SALM1 is unknown (Ko *et al*, 2006; Mah *et al*, 2010; Li *et al*, 2015;

1 Department of Clinical Genetics, Center for Neurogenomics and Cognitive Research, Amsterdam Neuroscience, VU University Amsterdam and VU Medical Center, Amsterdam, The Netherlands
2 Department of Functional Genomics, Center for Neurogenomics and Cognitive Research, Amsterdam Neuroscience, VU University Amsterdam and VU Medical Center, Amsterdam, The Netherlands
3 Department of Molecular and Cellular Neurobiology, Center for Neurogenomics and Cognitive Research, Amsterdam Neuroscience, VU University Amsterdam and VU Medical Center, Amsterdam, The Netherlands
*Corresponding author. Tel: +31 20 59 86946; E-mail: ruud.toonen@cncr.vu.nl
**Corresponding author. Tel: +31 20 59 86936; Fax: +31 20 598 6926; E-mail: matthijs@cncr.vu.nl

Lie et al, 2016). We show that SALM1 is present at pre- and postsynaptic membranes of mouse hippocampal neurons and that depletion of pre- or postsynaptic SALM1 impaired Neuroligin1- and Neurexin1β-mediated excitatory synapse formation and reduced synaptic vesicle clustering, synaptic transmission, and synaptic vesicle release. Furthermore, we discovered that SALM1 induced clustering of Neurexin in an F-actin- and PIP2-dependent manner. Together, our data suggest that SALM1 organizes synapse development by promoting F-actin/PIP2-dependent cis-oligomerization of Neurexin at the presynapse.

# Results

### SALM1 is a CASK/Mint1/Lin7b interactor in excitatory nerve terminals

To identify presynaptic CAMs that regulate synapse organization, we performed an immunoprecipitation mass spectrometry (IP-MS) proteomics screen using three components (CASK, Mint1, and Lin7b) of the proposed presynaptic organizer complex Munc18-1/CASK/Mint1/Lin7b (Butz et al, 1998) as baits. Proteins that were detected at least 10-fold higher as compared to the control (GluR2) IP-MS and detected using at least two of the CASK, Mint1, and Lin7b baits were considered putative interactors of the presynaptic

organizer complex (Fig 1A and B, Appendix Fig S1A). This resulted in 22 potential interactors including synaptic signaling proteins (e.g., Rab3GAP1 and Caskin1), cytoskeletal proteins (Actbl2 and Tubb4A), and liprin-α isoforms. The proteomics screen identified several known interactors of the CASK/Mint1/Lin7b complex (e.g., Caskin1, liprin-α) but did not detect other known interactors (e.g., Neurexins, Syncams, and Syndecans). The cell adhesion molecule SALM1, previously described as a postsynaptic protein (Lie et al, 2018), was the only adhesion molecule identified as a putative interactor of the CASK/Mint1/Lin7b presynaptic complex (Fig 1A and Appendix Fig S1A). In the reverse IP-MS, using SALM1 as bait, CASK was one of the most abundantly detected molecules (Fig 1A and B). Co-precipitation in HEK cells further confirmed the interaction between SALM1 and CASK (Fig 1C). Co-precipitation was not observed after truncation of SALM1, removing its PDZ binding domain (Fig 1C). Together, these data identify SALM1 as a novel binding partner of the CASK/Mint1/Lin7b presynaptic organizer complex, interacting directly with CASK, via its PDZ binding domain.

The interaction of SALM1 with the presynaptic organizer complex suggests that SALM1 is present in presynaptic terminals. To investigate the subcellular localization of SALM1, we generated a SALM1-specific antibody (Appendix Fig S1B). Multiple bands were detected in lysates of HEK cells expressing SALM1 (Appendix Figs S1B and S2). After Tunicamycin treatment, which blocks

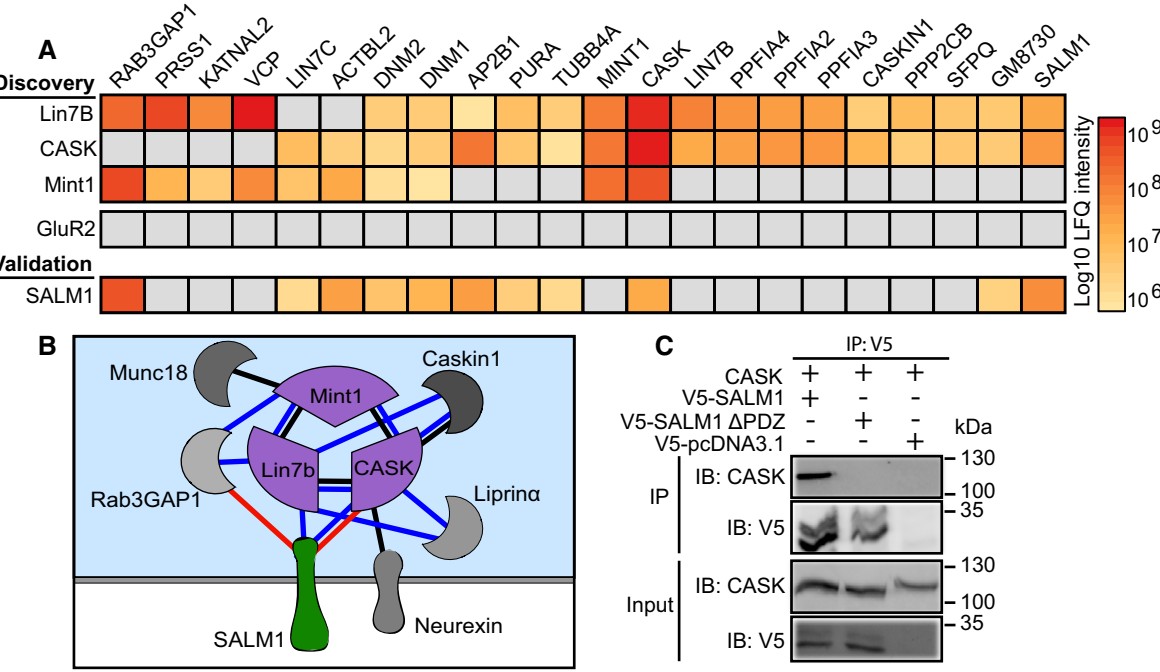

**Figure 1. SALM1 interacts with the presynaptic CASK/Mint1/Lin7b complex (related to Appendix Fig S1).**

A   Heat map showing 22 putative interactors of the Munc18-1/CASK/Mint1/Lin7b presynaptic complex identified in a proteomics screen using CASK, Mint1, or Lin7b as bait. Detection of the putative interactors using SALM1 or GluR2 (control) as bait is also shown. Bar values indicate average log10 LFQ intensity of three replicates. Gray indicates no detection in the IP.

B   Partial interactome of the putative SALM1/Munc18-1/CASK/Mint1/Lin7b complex identified by IP-MS analysis in (A). Blue lines indicate interactions identified in the IP-MS screen using CASK, Mint1, or Lin7b as bait. Red lines indicate interactions identified in reverse IP-MS with SALM1. Black lines indicate previously established interactions with the Munc18-1/CASK/Mint1/Lin7b complex (Butz et al, 1998; Tabuchi et al, 2002).

C   Co-immunoprecipitation of V5-tagged cytoplasmic SALM1, SALM1ΔPDZ, or an empty vector with CASK in HEK cells. The co-IP was repeated 3 times.

N-glycosylation, the top two bands of ~144 and ~146 kDa were lost (Appendix Fig S1C), suggesting that several glycosylated mature forms of SALM1 exist.

Staining for total (intracellular + surface) endogenous SALM1 using this new antibody revealed discrete puncta in (SMI-positive) axons and (MAP2-positive) dendrites of sandwich-cultured mouse hippocampal neurons at 16 days *in vitro* (DIV16, Fig 2A). SALM1 clusters highly overlapped (~90%) with excitatory synapse markers VGluT1 and Homer (Fig 2B). Higher magnification indicated that SALM1 puncta fully overlapped with presynaptic VGluT1 puncta and more partially with postsynaptic Homer puncta (Fig 2C). At an earlier developmental stage (DIV9), SALM1 distribution was similar to the distribution in DIV16 (Appendix Fig S3).

Immunoelectron microscopy was used to examine the subsynaptic localization of SALM1 in brain tissue sections. SALM1 was mostly detected in presynaptic terminals (~60%) of P75 mouse hippocampus (Fig 2D and E). SALM1 immunoreactivity was less abundantly detected (~40%) at postsynaptic terminals (Fig 2D and E). In line with being a binding partner for the presynaptic organizer complex CASK/Mint/Lin7b, SALM1 is preferentially localized to excitatory presynaptic terminals in mouse hippocampus.

## SALM1 depletion impairs Neurexin1β- and Neuroligin1-mediated synaptogenesis

Several cell adhesion molecules contain synaptogenic properties as indicated by the formation of hemisynapses by neurons on non-neuronal cells expressing an adhesion molecule of interest (e.g., Neurexins, Neuroligins) (Biederer & Scheiffele, 2007). Using this mixed culture assay, Mah *et al* showed that SALM1 does not induce presynapse formation when expressed in HEK cells co-cultured with primary neurons (Mah *et al*, 2010). We further investigated the synaptogenic properties of SALM1 by co-culturing sandwich-cultured mouse hippocampal neurons with HEK cells transfected with SALM1 tagged with extracellular pHluorin (SALM1-pHl). SALM1-pHl showed a punctate surface expression (Fig EV1A and B). In co-cultures, puncta of presynaptic marker VGluT1 or postsynaptic marker Homer were rarely detected on SALM1-pHl-positive HEK cells (Fig EV1C–F), corroborating the findings of Mah *et al* (2010). As expected, Homer or VGluT1 puncta were frequently detected on HEK cells expressing pHluorin-tagged Neurexin1β(-SS4) (Nrxn1β-pHl) or HA-tagged Neuroligin1AB (HA-Nlg1), respectively (Fig EV1C–F), as shown previously (Scheiffele *et al*, 2000; Graf *et al*, 2004).

To further investigate the role of SALM1 in synapse development, we developed two shRNA constructs (shRNA#1 and shRNA#2) directed against SALM1, which efficiently reduced endogenous SALM1 levels by ~50% (shRNA#1) or ~70-80% (shRNA#2) (Appendix Fig S2). HEK cells expressing Nrxn1β-pHl or HA-Nlg1 were co-cultured for 24 h with DIV9 sandwich-cultured mouse hippocampal neurons infected at DIV3 with shRNAs against SALM1 or scrambled shRNA (Fig 3A and F). Co-cultures of Nrxn1β-pHl expressing HEK cells with SALM1-depleted neurons showed a 2-fold reduction in the number of postsynaptic Homer puncta on Nrxn1β-pHl-positive HEK cells (Fig 3B and C). Endogenous SALM1 levels were rescued upon introduction of shRNA-resistant full-length SALM1 (rSALM1) or SALM1 lacking its PDZ binding domain (rSALM1ΔPDZ) (Appendix Fig S2). Introduction of rSALM1, but not rSALM1ΔPDZ, into SALM1-depleted neurons efficiently rescued the

number of Homer puncta on Nrxn1β-pHl-positive HEK cells (Fig 3B and C). The size and intensity of Homer puncta did not differ between conditions (Fig 3D and E).

Co-cultures of HEK cells expressing HA-Nlg1 with SALM1-depleted neurons showed a significant ~30% decrease in intensity and number of VGluT1 puncta on HA-Nlg1 expressing HEK cells while puncta size was unaffected (Fig 3G–J). The reduction in the number of VGluT1 puncta was rescued with rSALM1 but only partially with rSALM1ΔPDZ (Fig 3H). The reduced VGluT1 intensity was rescued with both rSALM1 and rSALM1ΔPDZ (Fig 3J). Together, these findings indicate that both pre- and postsynaptic SALM1 regulate synapse development via the Neurexin/Neuroligin synaptogenic pathway.

## SALM1 clusters Neurexin1β at the cell membrane in an F-actin- and PIP2-dependent manner

To investigate how SALM1 regulates the Neurexin/Neuroligin synaptogenic pathway, we studied the subcellular distribution of these three proteins in HEK cells. HEK cells were transfected with extracellularly tagged Nrxn1β-FLAG, HA-Nlg1AB, or SALM1-pHl in different combinations. When expressed alone, Nrxn1β-FLAG surface expression was largely diffuse (Fig 4A). Co-expression of Nrxn1β-FLAG and SALM1-pHl markedly altered the surface distribution of Nrxn1β-FLAG, inducing a ~65% increase in Nrxn1β-FLAG clusters at the surface (Figs 4B and 5J) and a ~45% decrease the Nrxn1β-FLAG diffusion ratio (Fig 5J) [D-ratio, calculated as the ratio of average intensity outside clusters (Appendix Fig S4C) over the average cluster intensity (Appendix Fig S4B)]. Surface SALM1-pHl puncta were found in close proximity to surface Nrxn1β-FLAG puncta (Fig 4B). HA-Nlg1AB showed punctate surface expression when expressed alone (Fig 4A). Co-expression of SALM1-pHl and HA-Nlg1AB did not alter HA-Nlg1AB membrane distribution, but surface SALM1-pHl and HA-Nlg1AB puncta were located in close proximity (Fig 4B). Hence, SALM1 clusters Neurexin1β at the surface of HEK cells.

To investigate how SALM1 clusters Nrxn1β, we tested a direct interaction using HEK cell co-precipitation. SALM1-pHl and Nrxn1β-FLAG did not co-precipitate, indicating that SALM1 does not directly interact with Nrxn1β (Appendix Fig S4A). Membrane proteins are often organized in microdomains enriched for specific membrane proteins, cytoskeletal components, and lipids. Since beta-neurexins interact with the actin cytoskeleton (Biederer & Sudhof, 2001), we tested the possibility that SALM1 organizes the submembrane cytoskeleton to indirectly cluster Neurexin in microdomains. HEK cells expressing SALM1-pHl or Nrxn1β-FLAG alone or in combination were stained for F-actin using fluorescently labeled Phalloidin. Control HEK cells showed a semi-diffuse subcortical F-actin network staining with some enrichment in microdomains (Appendix Fig S4D and E). HEK cells expressing Nrxn1β-FLAG alone showed diffuse Nrxn1β-FLAG surface localization with some minor enrichment at F-actin domains (Appendix Fig S4F and G), while more F-actin domains overlapped with surface SALM1-pHl puncta in HEK cells expressing SALM1-pHl alone (Appendix Fig S4H and I). In HEK cells co-expressing SALM1-pHl and Nrxn1β-FLAG, surface Nrxn1β-FLAG was clustered in puncta localized in close vicinity of SALM1/F-actin microdomains (Fig 5A and B). Inhibition of actin polymerization using Latrunculin A reduced the number of Nrxn1β-FLAG surface

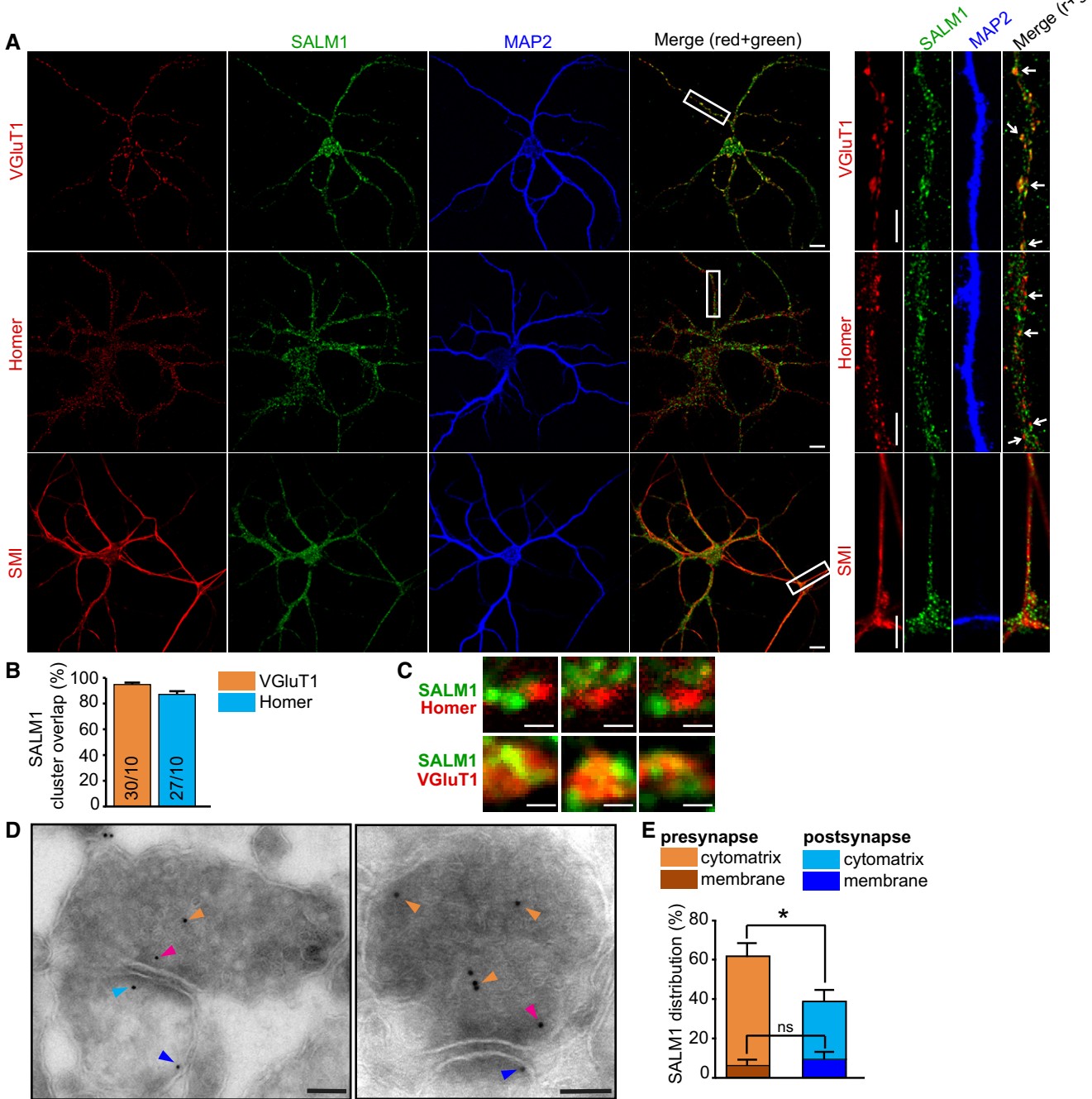

**Figure 2. SALM1 is localized to pre- and postsynapses of mouse excitatory hippocampal neurons (related to Appendix Figs S1–S3).**

A   Sandwich-cultured mouse hippocampal neurons stained at DIV16 for endogenous SALM1 (green), dendritic marker MAP2 (blue), and synapse markers Homer or VGluT1 (red), or the axonal marker SMI-312 (red). Boxes indicate area of zoom. Arrows indicate overlap between SALM1 and synapse markers. Bars = 10 μm in full neuron images. Bars = 5 μm in zoomed images.

B   Average overlap ± SEM of SALM1 clusters with VGluT1 or Homer puncta. Numbers in bars indicate number of zoomed images/total number of neurons in two independent experiments.

C   Example images showing differential overlap of SALM1 with Homer or VGluT1. Bars = 0.5 μm.

D   Electron micrographs showing subsynaptic localization of endogenous SALM1 in mouse hippocampal brain slices. SALM1 immunogold particles are detected in presynapses (orange and magenta arrows) and postsynapses (cyan and blue arrows). Bars = 100 nm.

E   Mean percentage of gold particles ± SEM detected in pre- versus postsynapses of mouse hippocampal slices stained for SALM1. Percentages are based on detected gold particles in 32 synapses in hippocampal brain slices of three different animals (Mann–Whitney *U*-tests with Bonferroni correction, ns = not significant, *$P$ < 0.025 after Bonferroni correction).

Source data are available online for this figure.

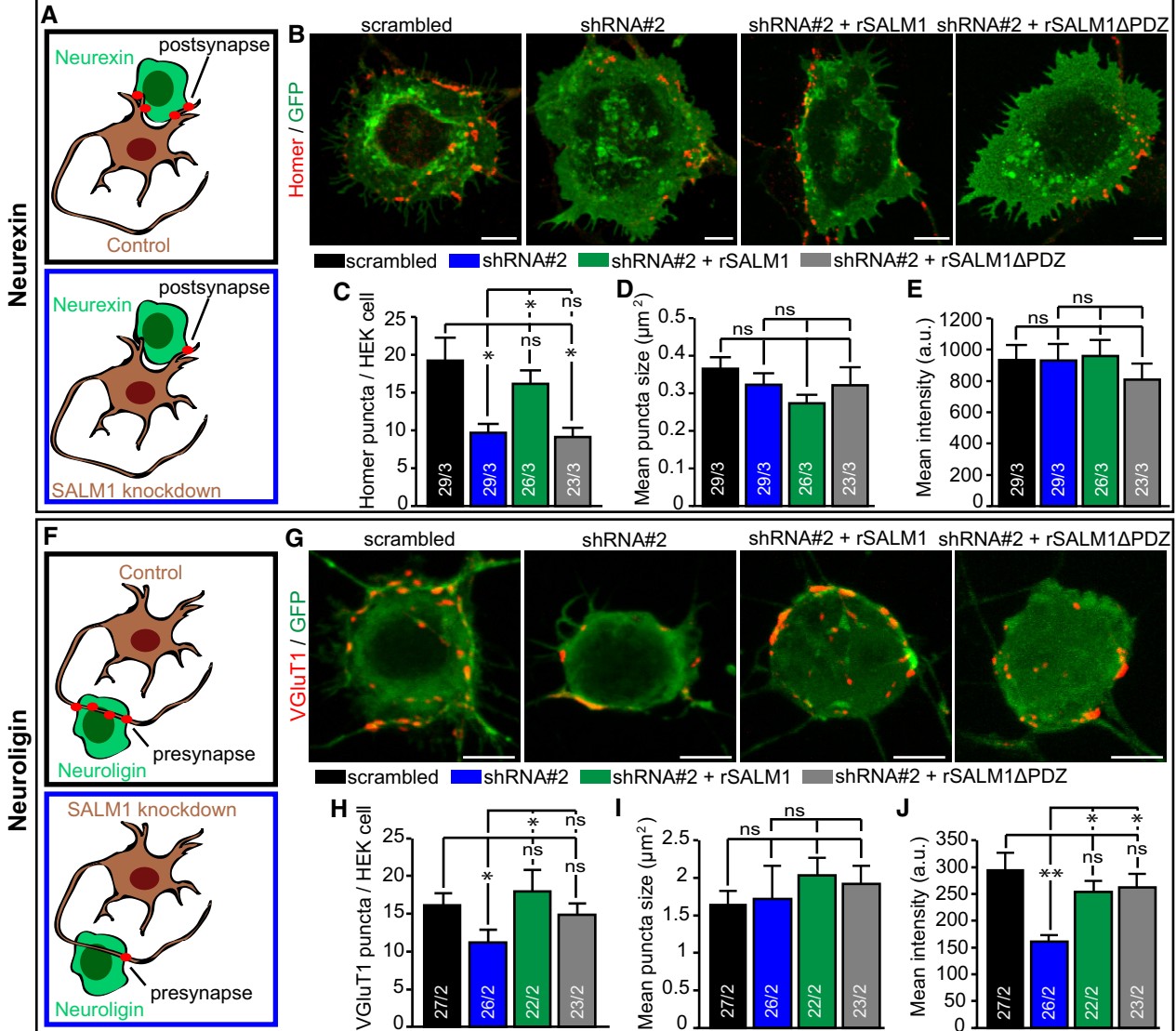

**Figure 3. SALM1 depletion reduces Neurexin- and Neuroligin-mediated synapse formation (related to Appendix Fig S2).**

A–J (A and F) Schematic representation of SALM1-depleted neurons forming postsynapses on Neurexin expressing HEK cells (A) or forming presynapses on Neuroligin expressing HEK cells (F). Example images showing postsynapses positive for postsynaptic marker Homer (red) formed on calcium phosphate transfected HEK cells expressing Nrxn1β-pHl (B) or showing presynapses positive for presynaptic vesicle marker VGluT1 (red) formed on calcium phosphate transfected HEK cells expressing HA-Nlg1 (G). HEK cells were co-cultured for 24 h with sandwich-cultured mouse hippocampal neurons infected with scrambled or SALM1 knockdown (shRNA#2) lentivirus (DIV3→10). Bars = 5 μm. Average number of Homer (C) or VGluT1 (H) particles ± SEM detected per HEK cell. Average size of the Homer (D) or VGluT1 (I) particles ± SEM detected per HEK cell. Average intensity ± SEM of Homer (E) and VGluT1 (J) puncta detected per HEK cell. For all graphs, the *n* is indicated in the bars and represents the total number of cells/total number of independent cultures. Kruskal–Wallis tests with post hoc paired comparisons were used for (C and H) ($P = 0.001$), (D) ($P = 0.065$), (E) ($P = 0.327$), (I) ($P = 0.06$), and (J) ($P = 0.003$). ns = not significant, *$P < 0.05$ and **$P < 0.01$.

Source data are available online for this figure.

clusters by ~75% and increased the D-ratio (~45%) in SALM1-pHl/Nrxn1β-FLAG expressing cells (Fig 5C, D, J and K). SALM1-pHl surface expression was unchanged upon Latrunculin A treatment (Fig 5C and D). Enhancement of F-actin polymerization using Jasplakinolide resulted in drastic increase in surface Nrxn1β-FLAG intensity in SALM1-pHl and Nrxn1β-FLAG co-expressing HEK cells (Appendix Fig S5A–E). Together, these findings indicate that SALM1-dependent Nrxn1β clustering depends on F-actin.

The interaction of beta-neurexins with F-actin depends on their C-terminal PDZ binding and the interaction with CASK (Hata *et al*, 1996; Butz *et al*, 1998). Since we found a similar interaction between SALM1 and CASK via the PDZ domain, we considered that this interaction might be important for the association of SALM1 and F-actin microdomains. To test this, HEK cells were transfected with SALM1ΔPDZ-pHl, Nrxn1β-FLAG, or both. SALM1ΔPDZ-pHl alone showed punctate surface expression similar to SALM1-pHl

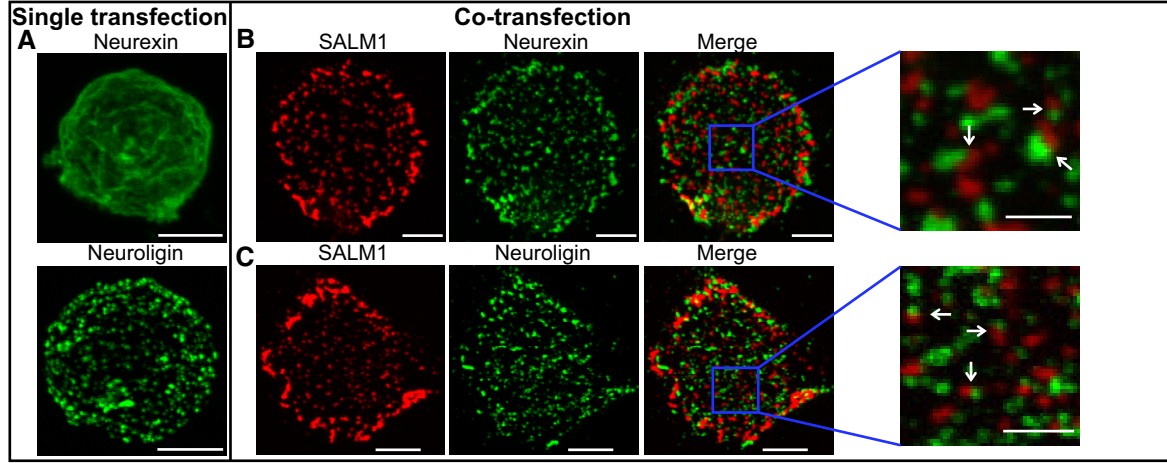

**Figure 4. SALM1 clusters Neurexin at the cell membrane of HEK cells.**

A  Examples of expression patterns of Nrxn1β-FLAG or HA-Nlg1 at the surface of calcium phosphate transfected HEK cells.
B  Example of surface expression patterns of SALM1-pHl (red) and Nrxn1β-FLAG (green) when co-expressed in calcium phosphate transfected HEK cells.
C  Example of surface expression patterns of SALM1-pHl (red) and HA-Nlg1 (green) when co-expressed in calcium phosphate transfected HEK cells.

Data information: Images represent collapsed z-stacks. Blue boxes indicate area of zoom; white arrows indicate examples of SALM1 puncta in close proximity of Nlg1 or Nrxn1β puncta. Bars = 5 μm in full HEK cell images. Bars = 2 μm in zoomed images.

(Appendix Fig S5H). Co-expression of SALM1ΔPDZ-pHl and Nrxn1β-FLAG resulted in punctate surface expression of Nrxn1β-FLAG similar to co-expression with SALM1-pHl (Appendix Fig S5I). Hence, SALM1 regulates F-actin-dependent clustering of Nrxn1β independently of its interaction with CASK.

Phosphatidylinositol 4,5-bisphosphate (PIP2) has been associated with microdomain organization (van den Bogaart et al, 2011) and the regulation of actin dynamics (Saarikangas et al, 2010; Chierico et al, 2014). We therefore hypothesized that PIP2 may play a role in F-actin-dependent clustering of Nrxn1β by SALM1. To visualize PIP2 in HEK cells, we co-transfected cells with the PH domain of Phospholipase C (PLC-PH-mCherry) (Milosevic et al, 2005). In control HEK cells, PIP2 microdomains often coincided with F-actin enrichments (Appendix Fig S4D and E), consistent with previous findings (Chierico et al, 2014). In cells expressing Nrxn1β-FLAG alone, surface Nrxn1β-FLAG was diffused with some minor enrichment in PIP2 microdomains (Appendix Fig S4F and G). Surface SALM1-pHl or SALM1ΔPDZ-pHl puncta strongly overlapped with PIP2 microdomains in SALM1-pHl or SALM1ΔPDZ-pHl expressing HEK cells (Appendix Figs S4H and I, and S5J and K). Co-expression of SALM1-pHl and Nrxn1β-FLAG resulted in clustering of surface Nrxn1β-FLAG and strong overlap between surface SALM1-pHl, Nrxn1β-FLAG, and PIP2 puncta (Fig 5F and G). Together, these findings indicate that SALM1 clusters Nrxn1β in PIP2 microdomains and is independent of SALM1's PDZ domain.

For other proteins, such as the membrane fusion protein Syntaxin1, PIP2 forms electrostatic interactions with polybasic amino acid subclusters in juxta membrane regions (van den Bogaart et al, 2011; Li et al, 2015). The five SALM isoforms also contain a highly conserved polybasic cluster directly adjacent to the transmembrane region and a relatively variable second polybasic cluster (Appendix Fig S6A). Therefore, we tested the ability of other SALM proteins to induce Nrxn1β clusters. SALM2, SALM3, SALM4, and

SALM5 all clustered Nrxn1β at the cell surface of HEK cells (Appendix Fig S6B). To test whether SALM1 clusters Nrxn1β via an electrostatic interaction with PIP2, we introduced two point mutations into the polybasic region of SALM1-pHl, changing charged Arginine and Lysine into Alanines (R556A and K558A; SALM1$^{RAKA}$-pHl, Fig 5E). SALM1$^{RAKA}$-pHl showed a punctate surface expression pattern on HEK cells similar to SALM1-pHl (Appendix Fig S4J and K). However, surface SALM1$^{RAKA}$-pHl puncta showed decreased overlap with PIP2 and F-actin microdomains (Appendix Fig S4J and K). SALM1$^{RAKA}$-pHl did not cluster Nrxn1β-FLAG at the cell membrane (Fig 5H–K). Furthermore, PIP2 depletion via expression of membrane-targeted Synaptojanin1 (Milosevic et al, 2005) resulted in diffuse Nrxn1β-FLAG surface expression in SALM1/Nrxn1β co-expressing HEK cells (Appendix Fig S5F and G). Together, these findings indicate that SALM1 clusters Nrxn1β via an electrostatic interaction of SALM1's polybasic domain with PIP2.

### SALM1 regulates Nrxn1β surface expression in mouse hippocampal neurons and enhances synaptogenesis

To determine whether SALM1 also regulates Nrxn1β distribution in neurons, we infected sandwich-cultured mouse hippocampal neurons at DIV3 with Nrxn1β-FLAG and scrambled, shRNA#2 (against SALM1), shRNA#2 + rSALM1 or shRNA#2 + rSALM1$^{RAKA}$ lentiviruses and analyzed cells 7 days later (DIV10). Surface Nrxn1β-FLAG intensity was reduced in VGluT1 positive synapses of SALM1-depleted neurons compared to control neurons (Fig 6A and B). Surface Nrxn1β-FLAG intensity was rescued by rSALM1, but not by rSALM1$^{RAKA}$ (Fig 6A and B).

To further investigate Nrxn1β clustering by SALM1 in neurons, we infected sandwich-cultured mouse hippocampal neurons at DIV3 with SALM1-pHl, SALM1$^{RAKA}$-pHl, SALM1ΔPDZ, and/or Nrxn1β-FLAG and analyzed cells 7 days later (DIV10). Neurons infected with

SALM1-pHl, SALM1RAKA-pHl, SALM1ΔPDZ, or Nrxn1β-FLAG alone showed punctate surface expression of SALM1-pHl, SALM1RAKA-pHl and SALM1ΔPDZ, and Nrxn1β-FLAG, respectively (Fig EV2A). SALM1RAKA-pHl and SALM1ΔPDZ surface/total ratio was decreased compared to SALM1-pHl (Fig EV2A and F). Co-expression of SALM1-pHl and Nrxn1β-FLAG drastically increased the number, size, and intensity of Nrxn1β-FLAG clusters at the cell surface (Fig EV2B–E). SALM1-pHl did not affect the total (intracellular + surface) expression levels of Nrxn1β-FLAG (Fig EV2G–J). Surface Nrxn1β-FLAG puncta largely overlapped with surface SALM1-pHl puncta (Fig EV2B). In contrast, co-expression of SALM1RAKA-pHl and

Nrxn1β-FLAG only modestly altered the number and size of surface Nrxn1β-FLAG puncta and did not alter Nrxn1β-FLAG puncta intensity (Fig EV2B–E). Co-expression of SALM1ΔPDZ and Nrxn1β-FLAG enhanced the number of surface Nrxn1β-FLAG puncta to similar extent as SALM1-pHl, while size and intensity of Nrxn1β-FLAG puncta were partially enhanced (Fig EV2B–E). SALM1RAKA-pHl and SALM1ΔPDZ-pHl did not affect total Nrxn1β-FLAG expression levels (Fig EV2G–J). Hence, SALM1 regulates the surface distribution of Nrxn1β in hippocampal neurons.

To determine if Nrxn1β-FLAG puncta recruited by SALM1-pHl participate in Neurexin/Neuroligin-mediated synaptogenesis, we

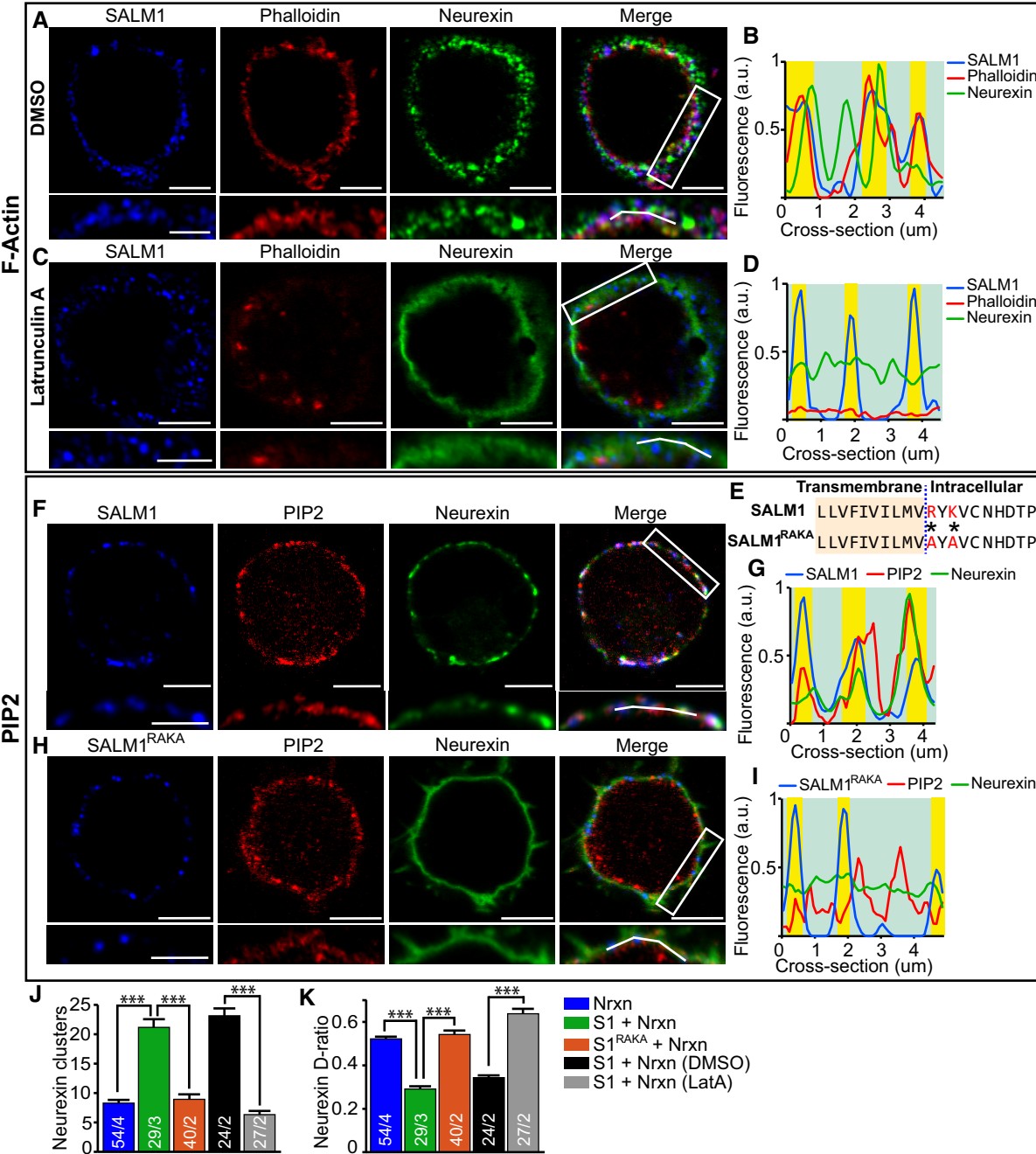

Figure 5.

**Figure 5.   Neurexin clustering by SALM1 is dependent on F-actin/PIP2 (related to Appendix Figs S4–S6).**

A–I   (A, C) Single z-slice images through calcium phosphate transfected HEK cells co-expressing SALM1-pHl and Nrxn1β-FLAG treated with DMSO (A) or Latrunculin A (C) and stained for surface GFP (blue), surface FLAG (green), and Phalloidin (red). White boxes indicate area of zoom; zoomed images are depicted below the full image for each channel. White lines in merged zoomed images represent cross sections used for intensity plots in (B, D). Bars = 5 μm in full HEK cell images. Bars = 3 μm in zoomed images. (B, D) Representative fluorescence intensity plots of cross sections depicted by white lines in the zoomed merge image in (A, C). SALM1 puncta are highlighted in yellow. (E) Partial amino acid sequence of the transmembrane and intracellular juxtamembrane domain of SALM1 and mutant SALM1^RAKA. The point mutations R556A and K558A are indicated by * and highlighted in red. Single z-slice images through HEK cells co-expressing calcium phosphate transfected SALM1-pHl (F) or SALM1^RAKA (H) (blue, surface staining) with Nrxn1β-FLAG (green, surface staining) and lentivirally expressed PLC-PH-mCherry (red). White boxes indicate area of zoom; zoomed images are depicted below the full image for each channel. White lines in merged zoomed images represent cross sections used for intensity plots in (B, D). Bars = 5 μm in full HEK cell images. Bars = 3 μm in zoomed images. (G, I) Fluorescence intensity plots of cross sections depicted by white lines in the zoomed merge image in (F, H). SALM1 puncta are highlighted in yellow.

J      Average number of surface Nrxn1β-FLAG clusters ± SEM per HEK cell for HEK cells expressing the different indicated constructs.

K      Average Nrxn1β-FLAG diffusion ratio (D-ratio) ± SEM per HEK cell for HEK cells expressing the different indicated constructs.

Data information: The n is indicated in the bars and represents the total number of cells/total number of independent cultures. S1 = SALM1-pHl, Nrxn = Nrxn1β-FLAG, and S1^RAKA = SALM1^RAKA-pHl. Kruskal–Wallis tests with post hoc paired comparisons were used on (J) (P < 0.001) and (K) (P < 0.001). ***P < 0.001.

Source data are available online for this figure.

co-cultured HA-Nlg1 expressing HEK cells for 24 h with sandwich-cultured hippocampal neurons infected at DIV3 with GFP (control), Nrxn1β-FLAG, SALM1-pHl, SALM1-pHl + Nrxn1β-FLAG, SALM1^RAKA-pHl + Nrxn1β-FLAG, or SALM1ΔPDZ-pHl + Nrxn1β-FLAG and analyzed cells at DIV10 (Fig 6C and D). Expression of Nrxn1β-FLAG or SALM1 alone did not alter the number, size, or intensity of presynaptic VGluT1 puncta formed on HA-Nlg1 expressing HEK cells compared to control condition (Fig 6D–G). In contrast, co-expression of SALM1-pHl and Nrxn1β-FLAG increased the number of VGluT1 puncta while size and intensity were unchanged (Fig 6D–G). Co-expression of SALM1^RAKA-pHl or SALM1ΔPDZ with Nrxn1β-FLAG did not alter the number, size, or intensity of presynaptic VGluT1 puncta (Fig 6D–G). Together, these findings indicate that Nrxn1β-FLAG clustering by SALM1 enhances Neurexin/Neuroligin-mediated synaptogenesis.

## Acute SALM1 depletion impairs synapse formation and synaptic vesicle clustering

To investigate the effects of pre- and postsynaptic SALM1 depletion on synapse development, single isolated mouse hippocampal neurons were infected with SALM1 shRNAs at three time points (DIV2, DIV7, and DIV9) to deplete total (pre- and postsynaptic) SALM1 levels and analyzed 7 days later (DIV9, DIV14, and DIV16, respectively) for cellular/synaptic morphology and synaptic function. Neurite length and branching was not altered upon depletion of SALM1 for all three time points (Appendix Fig S7A–F). However, SALM1 depletion significantly reduced the number of VGluT1-positive puncta when infected with SALM1 shRNAs at DIV2 and DIV7, but not at DIV9 (Figs 7A–E and EV3). A similar reduction was observed in the number of puncta for the postsynaptic marker Homer (Appendix Fig S7G–J) and the synaptic vesicle marker Synaptophysin1 (Appendix Fig S7N–P). The intensity of remaining VGluT1 and Synaptophysin1 puncta, but not of Homer puncta, was reduced (Figs 7D and E, and EV3, Appendix Fig S7K–P). The reduction in VGluT1 puncta number was rescued by rSALM1 expression, but not by rSALM1^RAKA or rSALM1ΔPDZ expression (Figs 7D and EV3). The reduced VGluT1 levels were also rescued by rSALM1 expression, but less efficiently by rSALM1ΔPDZ expression (Figs 7D and EV3). rSALM1^RAKA did not rescue VGluT1 levels (Figs 7D and EV3). Together, these data suggest that acute depletion of endogenous SALM1 in isolated primary neurons impairs synapse formation during the initial phases of synapse development, but not later. The

reduced staining for two synaptic vesicle markers in the remaining synapses suggests smaller vesicle clusters.

To further corroborate this conclusion, synapses of isolated primary neurons were analyzed after acute SALM1 depletion at the electron microscopy level. Morphometric analyses showed that the number of synaptic vesicles and the size of the vesicle cluster were reduced in synapses of SALM1-depleted neurons (Fig 7F–H), while active zone length, PSD length, or the number of membrane-proximal vesicles was not altered (Fig 7I–K). Together, these findings confirm that SALM1 is involved in synaptic accumulation of synaptic vesicles.

## SALM1 depletion reduces synaptic transmission and synaptic vesicle fusion

In addition, we investigated the effect of SALM1 depletion on synaptic transmission and synaptic vesicle fusion. SALM1 depletion in single isolated hippocampal neurons starting at DIV7 significantly reduced evoked postsynaptic current (EPSC) amplitude and spontaneous "mini" (mEPSC) frequency at DIV14, while mEPSC amplitude was unaltered (Fig 8A–E). Expression of rSALM1 or rSALM1^RAKA partially rescued the reduction in EPSC amplitude (Fig 8A and B). In contrast, SALM1 depletion starting at DIV9 did not significantly alter synaptic transmission at DIV 16 (Appendix Fig S8A–E).

Paired-pulse ratios (a measure for release probability) were unaltered between SALM1-depleted and control conditions (Fig 8F and Appendix Fig S8F). Similarly, normalized EPSC amplitudes during train stimulation (a measure for synaptic vesicle release/replenishment balance) were similar compared to control neurons (Fig 8G and H, Appendix Fig S8G). The cumulative peak amplitude and readily releasable pool (RRP) size were reduced in SALM1-depleted neurons (Fig 8I and J). The initial release probability was unaltered in SALM1-depleted neurons (Fig 8K). These findings suggest that SALM1 is not required for maintaining vesicle fusion/replenishment balance and release probability. The reduction in evoked responses upon SALM1 depletion may therefore be best explained by other factors such as the reduced number of synapses. The fact that mini amplitudes were unaffected by SALM1 depletion suggests that SALM1 is not required for normal postsynaptic receptor sensitivity in hippocampal neurons.

To investigate SV fusion upon acute SALM1 depletion, we used the SV protein Synaptophysin fused to pH-sensitive pHluorin

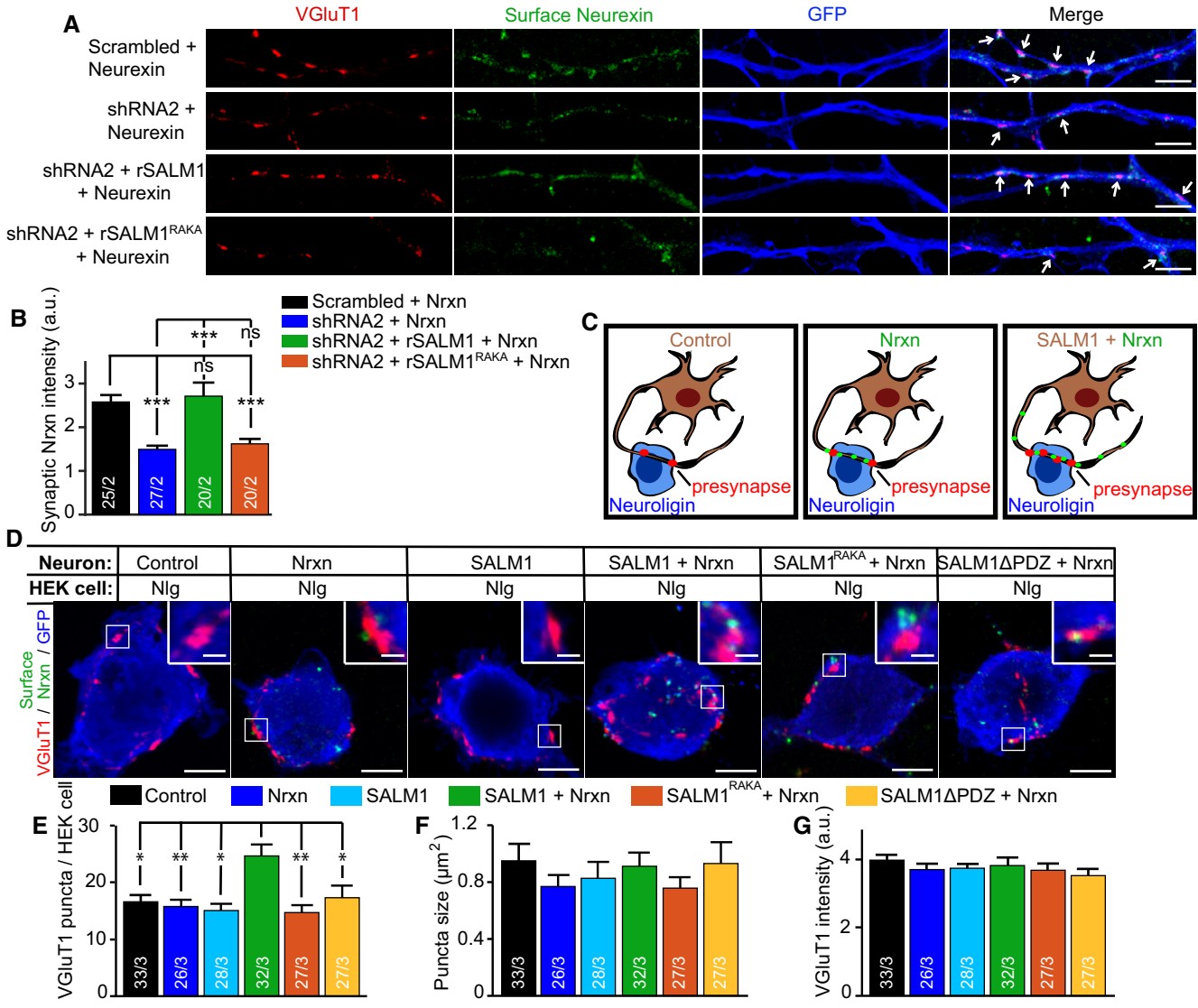

**Figure 6. SALM1 mediates synaptogenesis by clustering Neurexin in mouse hippocampal neurons.**

A Example images of neurites from DIV 10 sandwich-cultured mouse excitatory hippocampal neurons lentivirally infected at DIV4 with Nrxn1β-FLAG (green, surface staining) and scrambled, shRNA#2, shRNA#2+rSALM1, or shRNA#2+rSALM1[RAKA]. Arrows indicate overlap between VGluT1 puncta and Nrxn1β-FLAG clusters. Bars = 5μm

B Average surface Nrxn1β-FLAG intensity ± SEM detected in VGluT1 puncta per neuron.

C Schematic representation of neurons expressing GFP (control), Nrxn1βFLAG (green puncta), or SALM1-pHl with Nrxn1βFLAG forming presynapses (red puncta) on HA-Nlg1 expressing HEK cells.

D Example images showing presynapses positive for presynaptic marker VGluT1 (red) formed on calcium phosphate transfected HEK cells expressing HA-Nlg1 and stained for HEK cell filler GFP and surface Nrxn1β-FLAG (green). HEK cells were co-cultured for 24 h with sandwich-cultured mouse hippocampal neurons infected with GFP (control), Nrxn1β-FLAG, SALM1-pHl, SALM1-pHl+Nrxn1β-FLAG, SALM1[RAKA]-pHl+Nrxn1β-FLAG, or SALM1ΔPDZ-pHl+Nrxn1β-FLAG lentivirus (DIV3→10). Bars = 5 μm in full HEK cell images. Bars = 1 μm in zoomed images.

E–G Average number of VGluT1 puncta (E), puncta size (F), and puncta intensity (G) ± SEM detected per HEK cell.

Data information: For all graphs, the *n* is indicated in the bars and represents the total number of cells/total number of independent cultures. Kruskal–Wallis tests with post hoc paired comparisons were used on data sets (B) ($P < 0.001$), (E) ($P = 0.001$), and (F) ($P = 0.775$). A one-way ANOVA test was used in (G) ($P = 0.772$). ns = not significant, *$P < 0.05$, **$P < 0.01$, and ***$P < 0.001$.
Source data are available online for this figure.

(SypHy, Granseth *et al*, 2006). Single isolated neurons depleted for SALM1 at DIV7 showed a ~60% lower response to stimulation with 300 action potentials (300AP) at 10 Hz (which also induces release from the recycling/reserve vesicle pools) compared to control (F300, Fig 8L and O). The fluorescence response to brief application of $NH_4^+$, which neutralizes the luminal pH of SVs and reveals the total SypHy labeled vesicle pool, was ~60% lower in SALM1-depleted neurons (Fmax, Fig 8L and N). The F300 and Fmax

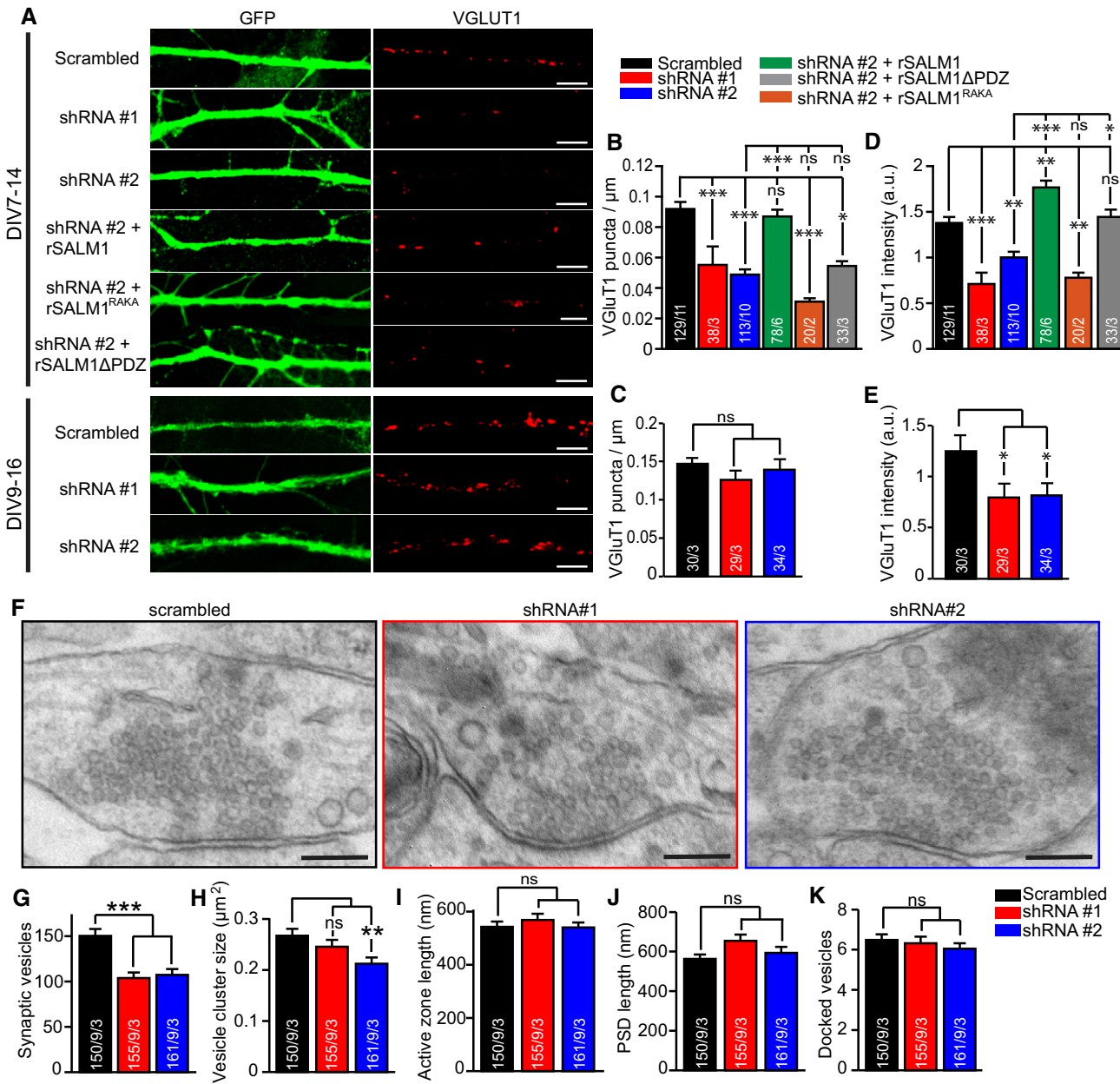

**Figure 7. Knockdown of SALM1 decreases synapse formation and synaptic vesicle clustering (related to Fig EV3 and Appendix Fig S7).**

A  Example images of neurites from single isolated (autaptic) mouse excitatory hippocampal neurons stained for GFP and VGluT1. Cells were lentivirally infected with scrambled or SALM1 shRNA constructs at different time points (DIV7 or DIV9) and analyzed 7 days later (DIV14 or DIV16). At DIV7, cells were lentivirally infected with shRNA#2 + rSALM1, shRNA#2 + rSALM1^RAKA or with shRNA#2 + rSALM1ΔPDZ. Bars = 5 μm.

B, C  Average number of VGluT1 puncta per μm neurite ± SEM per neuron for DIV7→14 (B) and DIV9→16 (C).

D, E  Average intensity ± SEM of VGluT1 puncta per neuron for DIV7→14 (D) and DIV9→16 (E).

F  Electron micrographs of synapses from DIV14 autaptic hippocampal neurons lentivirally infected at DIV7 with scrambled, shRNA#1, or shRNA#2 (black, red and blue boxed images, respectively). Synaptic clefts are indicated by green arrows. Presynapses are localized above the synaptic cleft showing distinct vesicular structures; postsynapses are localized below the synaptic cleft. Bars = 100 nm.

G  Average number of synaptic vesicles per synapse ± SEM in electron micrographs.

H  Average size of the synaptic vesicle cluster per synapse ± SEM.

I, J  Average length of the presynaptic active zone (K) and the postsynaptic density (L) ± SEM per synapse.

K  Average number of membrane proximate vesicles ± SEM per synapse.

Data information: For all graphs, numbers in bars indicate total number of neurons/total number of independent cultures. Kruskal–Wallis tests with post hoc paired comparisons were used for all data sets, $P < 0.001$ for (B, D, and G), $P = 0.004$ for (H), $P = 0.349$ for (C), $P = 0.031$ for (E), $P = 0.879$ for (I), $P = 0.368$ for (J), and $P = 0.463$ for (K). ns = not significant, $*P < 0.05$, $**P < 0.01$, and $***P < 0.001$.

Source data are available online for this figure.

fluorescence was efficiently rescued by rSALM1 and rSALM1ΔPDZ, but not by rSALM1$^{RAKA}$ (Fig 8R, T and U). The ratio of fluorescence increase upon stimulation divided by the maximum fluorescence increase upon $NH_4^+$ superfusion was unaltered upon SALM1 depletion or rescue with rSALM1, rSALM1$^{RAKA}$, or SALM1ΔPDZ (Fig 8M, P, S and V). Finally, SALM1 depletion did not affect the percentage of non-responding (silent) synapses (Fig 8Q). The data suggest that the reduction in vesicle fusion is fully explained by the reduction in vesicle pool size. Together with the electrophysiology data, this shows that SALM1 controls vesicle fusion by regulating vesicle pool size, but not by regulating release probability or vesicle release/replenishment balance. It is likely that the reduction in synapse numbers underlies the reduction in evoked EPSC response and miniature frequency upon SALM1 depletion.

## Discussion

In this study, we identify a molecular mechanism of excitatory synapse development via the adhesion molecule SALM1, which interacts with the presynaptic organizer complex Munc18-1/CASK/Mint1/Lin7b and clusters Neurexin in F-actin- and PIP2-rich membrane microdomains.

### SALM1 preferentially localizes to presynaptic terminals of hippocampal neurons

Three lines of evidence, based on protein–protein interactions, immunocytochemistry, and immunoelectron microscopy, indicate that endogenous SALM1 accumulates in pre- and postsynaptic

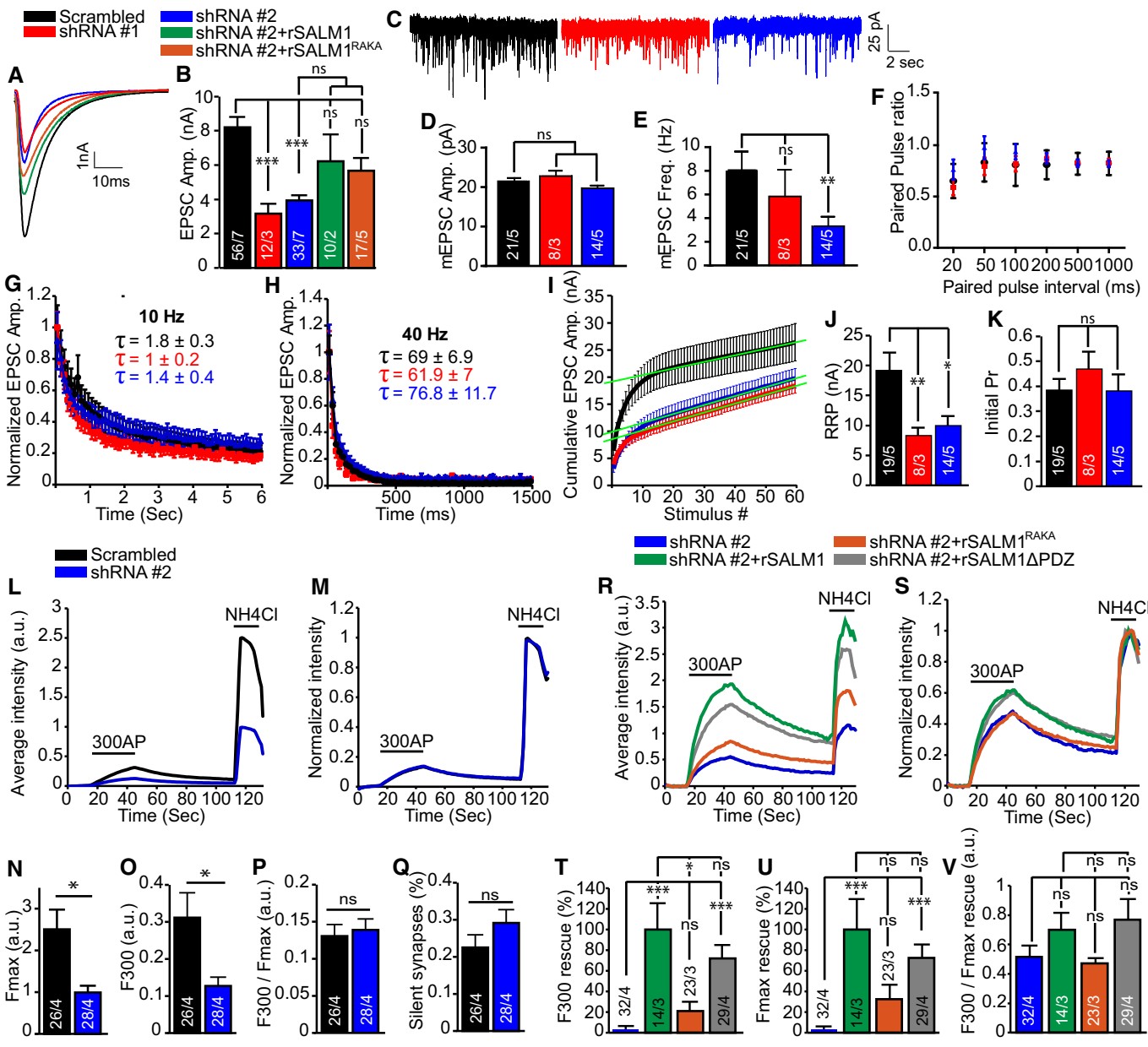

**Figure 8.**

**Figure 8.  SALM1 depletion during early developmental stage reduces synaptic transmission and synaptic vesicle fusion (related to Appendix Fig S8).**

Patch-clamp analysis (A–K) on single isolated mouse hippocampal neurons lentivirally infected with the indicated constructs at DIV7 and analyzed at DIV14-15.

A   Example traces of EPSCs in control (black), shRNA#1 (red), shRNA#2 (blue), shRNA#2+rSALM1 (green), or shRNA#2+rSALM1$^{RAKA}$ (orange) expressing cells.

B   Average evoked EPSC amplitudes $\pm$ SEM. Kruskal–Wallis ($P < 0.001$). These experiments could not be performed with visual confirmation of lentiviral infection of rescue constructs, which typically is around 80%. Together with the high variability in evoked synaptic responses, this uncertain factor may explain that the average EPSC amplitudes are incompletely restored after SALM1 knockdown and that the RAKA mutant is not significantly different from the WT.

C   Example traces of mEPSCs from control (black), shRNA#1 (red), or shRNA#2 (blue) expressing cells.

D, E   Average mEPSC amplitudes $\pm$ SEM (D) or average frequencies $\pm$ SEM (E). Kruskal–Wallis ($P < 0.001$ for E and $P = 0.879$ for D).

F   Average paired-pulse ratios obtained using different intervals $\pm$ SEM. $n = 20$, $n = 9$, and $n = 12$ for scrambled, shRNA#1, and shRNA#2, respectively.

G   Average normalized EPSC response $\pm$ SEM upon 10 Hz train stimulation. $n = 20$, $n = 9$, and $n = 13$ for scrambled, shRNA#1, and shRNA#2, respectively. Tau $\pm$ SEM is indicated in the graph for each condition and was not significant (Kruskal–Wallis, $P = 0.084$).

H   Average normalized EPSC response $\pm$ SEM upon 40 Hz train stimulation. $n = 20$, $n = 9$, and $n = 13$ for scrambled, shRNA#1, and shRNA#2, respectively. Tau $\pm$ SEM is indicated in the graph for each condition and was not significant (Kruskal–Wallis, $P = 0.897$).

I   Cumulative EPSC responses to 40 Hz train stimulation. Green lines represent extrapolation used to determine RRP size.

J   Average RRP size $\pm$ SEM determined by back extrapolation of cumulative EPSCs in I (one-way ANOVA with post hoc Games-Howell, $P = 0.007$).

K   Average initial release probability (Pr) $\pm$ SEM per neuron calculated as the ratio of EPSC$_0$/RRP size (Kruskal–Wallis, $P = 0.347$).

L   Average trace of the fluorescence intensity of SypHy in calcium phosphate transfected isolated hippocampal neurons expressing scrambled shRNA (black) or shRNA#2 (blue). The timing of the 300 action potentials at 10 Hz stimulus and the exposure to NH$_4$Cl are indicated in the graph.

M   Average SypHy intensity trace normalized to the maximum intensity upon NH$_4$Cl superfusion for each of the given conditions.

N, O   Average SypHy intensity $\pm$ SEM upon NH$_4$Cl exposure (N) or upon stimulation with 300 action potentials (O) (Mann–Whitney $U$-test, $P = 0.02$ for M and $P = 0.03$ for N).

P   Normalized average SypHy intensity $\pm$ SEM upon 300 action potentials stimulation (Mann–Whitney $U$-test, $P = 0.899$).

Q   Average percentage of silent synapses $\pm$ SEM (Mann–Whitney $U$-test, $P = 0.652$).

R   Average trace of the fluorescence intensity of SypHy in lentivirally infected isolated hippocampal neurons expressing shRNA#2 (blue), shRNA#2+rSALM1 (green), shRNA#2+SALM1$^{RAKA}$ (orange), or shRNA#2+SALM1$\Delta$PDZ (gray). The timing of the 300 action potentials at 10 Hz stimulus and the exposure to NH$_4$Cl are indicated in the graph.

S   Average SypHy intensity trace normalized to the maximum intensity upon NH$_4$Cl superfusion for each of the given conditions in (Q).

T, U   Average percentage $\pm$ SEM rescue compared to rSALM1 upon NH$_4$Cl exposure (U) or upon stimulation with 300 action potentials (T) (Kruskal–Wallis, $P < 0.001$ for T and U).

V   Normalized average SypHy intensity $\pm$ SEM upon 300 action potentials stimulation (Kruskal–Wallis, $P = 0.06$).

Data information: For all graphs, numbers in bars indicate total number of cells/total number of independent experiments. ns = not significant, *$P < 0.05$, **$P < 0.01$ and ***$P < 0.001$.

Source data are available online for this figure.

---

locations of hippocampal neurons and is most abundant on the presynaptic side. This preferential localization to the presynaptic side has not been described for SALM proteins before. SALM4 localizes to both pre- and postsynaptic terminals of hippocampal neurons and has been implicated in postsynaptic functions (Seabold *et al*, 2008), and the subsynaptic localization of SALM2, SALM3, and SALM5 is unknown. Hence, SALMs are not only postsynaptic proteins, as previously proposed (Lie *et al*, 2018), but may also have presynaptic functions. Immunoelectron microscopy detected only a fraction of endogenous SALM1 molecules on the synaptic membrane and the majority in the presynaptic cytomatrix. Surface levels of SALM1 may be dynamically regulated between cytosolic and membrane pools.

## SALM1 mediates synapse development

Upon total (pre- and postsynaptic) SALM1 depletion in single isolated neurons, the number of synapses formed was severely reduced and synaptic transmission was impaired to a similar extent. These observations suggest that SALM1 is a major factor in synapse formation and/or maintenance in hippocampal neurons. SALM1 depletion during the first postnatal week produced a strong effect on synapse density (~50% reduction), but less so during the second. This is consistent with the fact that SALM1 is highly expressed during the first postnatal week (Wang *et al*, 2006). This suggests that SALM1 promotes the formation of new synapses, but may not be necessary for maintenance of existing synapses. Between the first and second postnatal week, the expression of many synaptic proteins increases (Petralia *et al*, 2005; Ko *et al*,

2006) and SALM1 function may become redundant during later developmental stages.

Remaining synapses in SALM1-depleted neurons showed a normal release probability and normal synaptic plasticity, e.g., normal rundown kinetics and vesicle release/replenishment balance during repetitive stimulation. This suggests that presynaptic calcium dynamics and the regulation of synaptic vesicle fusion and replenishment in these synapses are unaffected by SALM1 depletion. This may indicate that a subpopulation of synapses forms and functions independently of SALM1. However, immunocytochemistry and electron microscopy indicated a reduced synaptic vesicle cluster in these synapses. This indicates that also in the remaining synapses, SALM1 has an additional role in clustering synaptic vesicles, but not in the fusion and replenishment of the subpool involved in synaptic transmission [20–40% of the total pool, see Fernandez-Alfonso and Ryan (2006)]. The interaction of SALM1 with the presynaptic organizer complex Munc18-1/CASK/Mint1/Lin7b (Butz *et al*, 1998) discovered in this study may be involved in this aspect of SALM1 function. In addition, we found that SALM1 regulates the surface clustering of Neurexin, another interactor of the Munc18-1/CASK/Mint1/Lin7b complex, which also regulates synaptic vesicle clustering (Hata *et al*, 1996; Butz *et al*, 1998; Dean *et al*, 2003). Hence, SALM1 may accumulate synaptic vesicles in developing synapses both by interacting with the Munc18-1/CASK/Mint1/Lin7b complex via its PDZ binding domain and by clustering Neurexin.

In addition to clustering synaptic vesicles, the interaction of SALM1 with the Munc18-1/CASK/Mint1/Lin7b complex via its PDZ binding domain may help to stabilize SALM1 at the synaptic

membrane. This is supported by previous findings that deletion of the PDZ binding domain of SALM1 was previously shown to impair SALM1 surface expression (Seabold et al, 2012) and our findings that even though SALM1ΔPDZ clusters Neurexin on HEK cell membranes, SALM1ΔPDZ rescue upon SALM1 depletion in neurons was impaired.

The effects of SALM1 depletion on synapse formation were strong relative to previously published studies on depletion of CAMs (Robbins et al, 2010; Yim et al, 2013; Chen et al, 2017; Jiang et al, 2017). This suggests that SALM1 is a major regulator of these processes. In addition, the method of CAM depletion may also influence the strength of the observed phenotype. Constitutive KO models often lead to milder phenotypes compared to acute CAM depletion using RNAi-mediated knockdown or conditional KO models (Chih et al, 2005; Chubykin et al, 2007; Etherton et al, 2009; de Wit et al, 2009). Consistently, two independent studies have recently reported milder phenotypes for SALM1 constitutive KO mice compared to our acute, RNAi-mediated SALM1 knockdown method (Morimura et al, 2017; Li et al, 2018). These studies found little to no effect on hippocampal excitatory synaptic density in SALM1 constitutive KO models. However, both constitutive KO studies report rather dissimilar behavioral and synaptic phenotypes, even though both studies generated constitutive SALM1 KO mice using similar approaches. This suggests complex and variable phenotypic buffering (Sudhof, 2017). Recent major discoveries on CAM function are based on acute approaches (Zhang et al, 2015; Chen et al, 2017; Schroeder et al, 2018).

## SALMs control synapse development by clustering synaptic proteins

At the postsynapse, SALMs differentially interact with NMDA and AMPA receptors (SALM1 and SALM2) (Wang et al, 2006), scaffolding protein PSD95 (SALM1-3) (Ko et al, 2006; Morimura et al, 2006; Wang et al, 2006), and trans-interact with presynaptic LAR-PTP's (SALM2, SALM3, and SALM5) (Li et al, 2015; Choi et al, 2016; Goto-Ito et al, 2018). In addition, SALM1, SALM2, SALM3, and SALM5 depletion decreases synaptic density, while SALM4 depletion increases synaptic density [our data and Ko et al (2006), Mah et al (2010), Li et al (2015), Lie et al (2016)]. Furthermore, SALMs have differential spatiotemporal distributions, e.g., with SALM1 highly expressed during early development, while SALM2 expression increases during late development (Ko et al, 2006; Wang et al, 2006). The differential interaction partners and spatiotemporal expression of individual SALMs suggest that different SALMs regulate synapse development through distinct mechanisms which may also explain the lack of redundancy upon SALM1 depletion. However, for all the interaction partners mentioned above, the main role of SALMs appears to be the clustering of these proteins (Ko et al, 2006; Wang et al, 2006; Mah et al, 2010; Goto-Ito et al, 2018; Lin et al, 2018), indicating a common role for SALMs in controlling synapse assembly by clustering of synaptic proteins. Our finding that all five SALMs cluster Neurexin in heterologous cells further supports such a common role. Together, the picture emerges that different SALMs control synapse development by clustering specific subsets of synaptic proteins.

## SALM1 cis-oligomerizes Neurexin via F-actin and PIP2

In this study, we found that SALM1 homomeric clusters were sufficient to induce clustering of Neurexin in an F-actin/PIP2-dependent manner, and that SALM1 also increased Neurexin clustering on the surface of hippocampal neurons without affecting total Neurexin levels. Our data suggest that SALM1 indirectly cis-oligomerizes Neurexin according to the following model (Fig 9):

1. SALMs form homo- and heteromeric cis-interactions via their extracellular domains, as previously observed (Seabold et al, 2008) and supported by our observations that (i) SALM1 forms clusters on the membrane of HEK cells and neurons; (ii) that SALM1 oligomers remained stable upon disruption of the Actin cytoskeleton and upon deletion or mutation of the PDZ binding domain or the polybasic domain of SALM1. In addition, the ability of SALM1 to form heteromeric cis-interactions suggests that endogenous SALM1 oligomers may be heteromeric, containing other CAMs with potential synaptogenic properties.

2. The clustering of multiple polybasic domains of SALM1 generates a local increase in positive charge that recruits negatively charged PIP2 molecules. This is supported by previous findings that PIP2 clusters on the cell membrane via electrostatic interactions with polybasic domains of membrane proteins (van den Bogaart et al, 2011) and our findings that (i) PIP2 was highly enriched at SALM1 microdomains; and (ii) the polybasic mutant SALM1$^{RAKA}$ fails to enrich PIP2.

3. SALM1 induces local enrichments in F-actin via PIP2. Previous studies showed that PIP2 microdomains reciprocally regulate F-actin dynamics by recruiting F-actin nucleation complexes (Papayannopoulos et al, 2005; Chierico et al, 2014). We furthermore found F-actin enrichments at SALM1 clusters, but not at SALM1$^{RAKA}$ clusters. The inability of SALM1$^{RAKA}$ to recruit F-actin may be an indirect result of the inability to recruit PIP2.

4. Nrxn1β is recruited to SALM1 microdomains via interactions with F-actin. Nrxn1β interacts with F-actin via scaffolding protein CASK and protein 4.1 (Biederer & Sudhof, 2001). In addition, we found that Nrxn1β oligomerization by SALM1 was dependent on F-actin and PIP2.

In contrast to trans-interactions, little is known about the complexity of cis-interactions between CAMs and their function in synapse assembly. Few studies have shown that direct homomeric cis-interactions between CAMs and subsequent oligomerization promotes trans-interactions and consequently affects synapse formation and function (Fogel et al, 2011; Shipman & Nicoll, 2012; Lin et al, 2018). Our model suggests a principle of indirect F-actin/PIP2-dependent cis-oligomerization of Neurexin by SALM1 through a mechanism that requires (i) SALM1 oligomerization through direct cis-interactions and (ii) the presence of a juxtamembrane polybasic domain in SALM1. Polybasic domains are commonly observed in neuronal (e.g., SynCAM1, LRRTM2, PTPRD) and non-neuronal (e.g., BMPR1B, ICAM1, VCAM1) CAMs in the UniProt protein database (The UniProt Consortium, 2017). This suggests that oligomerization of CAMs via indirect F-actin/PIP2-dependent cis-interactions is a common mechanism that may orchestrate supra-molecular organizations in neuronal as well as non-neuronal cells. In addition, we showed that presynaptic SALM1 depletion

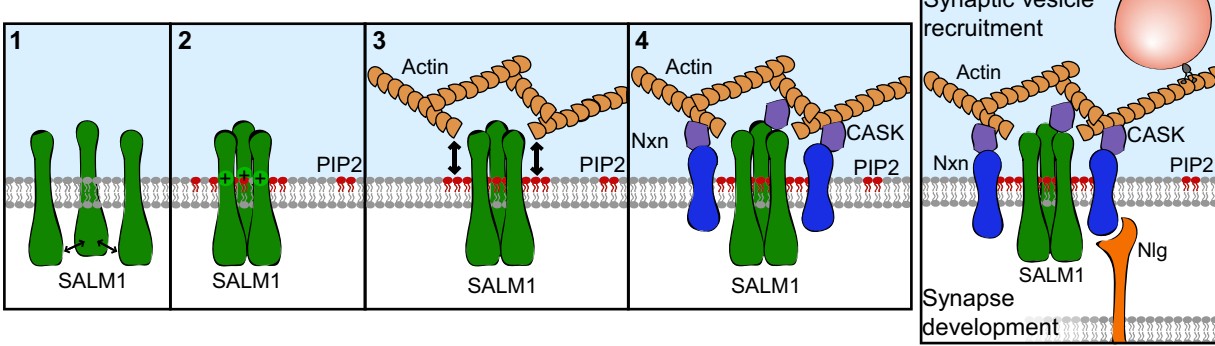

**Figure 9. SALM1-dependent Neurexin clustering model.**

Schematic model illustrating clustering of Neurexin by SALM1 in four steps. (1) SALM1 self-clusters on cell membranes through direct homomeric *cis*-interactions. (2) SALM1 clusters recruit negatively charged PIP2 via electrostatic interactions with SALM1's polybasic domain. (3) PIP2 microdomains formed by SALM1 induce local F-actin network formation. (4) Neurexin is recruited to SALM1 microdomains via F-actin and PIP2. Together, the data in this study suggest a role for the Neurexin/SALM1/PIP2/F-actin complex in synapse development and synaptic vesicle recruitment.

impaired Neuroligin induced synapse formation, suggesting that indirect *cis*-oligomerization of Neurexin by SALM1 is upstream of Neurexin/Neuroligin *trans*-interactions and subsequent synaptogenesis.

# Materials and Methods

### Contact for reagent and resource sharing

Further information and requests for resources and reagents should be directed to and will be fulfilled by the Lead Contact, Matthijs Verhage (matthijs@cncr.vu.nl).

### Experimental model and subject details

#### Animals

Wild-type E18 mouse embryos were obtained by cesarean section of pregnant female C57/Bl6 mice. Newborn P0-P1 pups from pregnant female Wistar rats were used for glia preparations. All animal experiments were performed according to the Dutch legislations for the use of laboratory animals.

#### HEK cell cultures

Human Embryonic Kidney 293T cells (HEK cells) were cultured in DMEM/F12 medium with L-glutamine supplemented with 10% FCS, 1% NEAA, and 1% Pen/Strep (all Gibco). Cells were plated in 6-well culture plates (Greiner) at equal densities 1 day prior to transfection at 37°C, 5% $CO_2$.

### Method details

#### Proteomics

P2+microsome fractions from adult mice were solubilized with 1% detergent extraction buffer (1% n-Dodecyl β-D-maltoside (DDM), 150 mM NaCl, 25 mM HEPES, pH 7.4, and protease inhibitor (Roche)). Extracts were incubated with antibodies against CASK, Lin7b, Mint1, SALM1, or GRIA2 at 4°C overnight on a mechanical

rotator. 50 μl of protein A/G PLUS-Agarose beads (Santa Cruz) was washed four times with washing buffer (0.1% DDM, 150 mM NaCl, 25 mM HEPES, pH 7.4) before it was added to the samples for 1 h at 4°C. The buffer was completely removed using an insulin syringe before storing the samples at −20°C until further use.

An SDS–PAGE LC-MS/MS approach was used for protein identification as described previously (Chen *et al*, 2011). In short, after separation on the SDS–PAGE gel proteins were trypsin digested. The resulting peptides were separated on a capillary C18 column using a nano LC-ultra 1D plus HPLC system (Eksigent) and analyzed online with an electrospray LTQ-Orbitrap Discovery mass spectrometer (Thermo Fisher Scientific). MS-MS spectra were searched against the UniProt proteomics database (version 2013-01-06) with MaxQuant software (version 1.3.0.5). Methionine oxidation and protein N-terminal acetylation were set as variable modifications, and cysteine alkylation with acrylamide was set as fixed modification. The maximum mass deviations of parent and fragment ions were set to 6 ppm and 0.5 Da, respectively. Trypsin was chosen as the digestion enzyme, and the maximum missed cleavage was set at 2. Each valid protein hit should contain at least one unique peptide. The false discovery rates of both peptides and proteins were set within a threshold value of 0.01. The MaxLFQ algorithm in MaxQuant was used to normalize the data.

#### Dissociated hippocampal cultures

Isolated single neurons were grown on glia microislands as described previously (Wierda *et al*, 2007). Glass coverslips (Menzel) were etched in 1M HCl for at least 2 h and neutralized with 1M NaOH for maximum 1 h, washed thoroughly with MilliQ water, and washed once with 70% ethanol. Coverslips were stored in 96% ethanol and coated with agarose type II-A (0.0015% in $H_2O$, Sigma) prior to microdot application. Coating was done by spreading a thin layer of agarose solution (heated in microwave and kept at 55°C during use) with a cotton swab over the entire coverslip. For microisland plates, microdots were created using a custom made rubber stamp (dot diameter 250 μm) to apply solution consisting of 0.5 mg/ml Poly-D-Lysine (Sigma), 3.66 mg/ml collagen (BD biosciences), and 17 mM acetic acid (Sigma) by stamping from a wet filter paper (3 mm

cellulose chromatography paper (Whatman)). For continental and sandwich cultures, coverslips or plates were sprayed with the same Poly-D-Lysine solution using an airbrush. Coverslips were UV-sterilized for 20 min before further use. Astrocytes were plated at 6–8 K/well for microislands and 25K/well for continental/sandwich cultures in pre-warmed DMEM medium supplemented with 10% FCS, 1% NEAA, and 1% Pen/Strep (all Gibco). After 4–5 days, DMEM medium was replaced by Neurobasal medium supplemented with 2% B-27, 1.8% HEPES, 0.25% glutamax, and 0.1% Pen/Strep (all Gibco) and neurons were added to the cultures.

Hippocampus and cortex were dissected from embryonic day 18 (E18) wild-type C57/Bl6 mice and collected in ice-cold Hanks Buffered Salt Solution (HBSS; Sigma), buffered with 7 mM HEPES (Invitrogen). Tissues were incubated in Hanks-HEPES with 0.25% trypsin (Invitrogen) for 20 min at 37°C. After washing, neurons were triturated using a fire-polished Pasteur pipette and counted in a Fuchs-Rosenthal chamber. The cells were plated in Neurobasal medium supplemented with 2% B-27, 1.8% HEPES, 0.25% glutamax, and 0.1% Pen/Strep (all Gibco) on 18 mm glass coverslips. For microisland cultures, neurons were plated at densities between 1,000 and 2,000 neurons per well. For continental cultures, neurons were plated at a density of 25K per well.

To make sandwich cultures, flattened tweezers were heated and shortly placed on the bottom of wells of 12-well plastic culture plates (Greiner) to create small extrusions. Glia were plated on the bottom of the wells as described above. Glass coverslips (18 mm) were washed in 96% ethanol and flamed dry. Coverslips were incubated with sterile $H_2O$ (Baxter) containing 2.5 μg/ml Laminin (Sigma) and 0.0005% Poly-L-Ornithine (Sigma) for 2 h at 37°C. Coverslips were then washed three times with sterile $H_2O$ (Baxter), and Neurobasal medium supplemented with 2% B-27, 1.8% HEPES, 0.25% glutamax, and 0.1% Pen/Strep (all Gibco) was added. Neurons were plated at a density of 15K per well and allowed to attach to the coverslip for 24 h. DMEM medium on glia cultures was then replaced for Neurobasal medium supplemented with 2% B-27, 1.8% HEPES, 0.25% glutamax, and 0.1% Pen/Strep (all Gibco). Glass coverslips containing the neurons were then placed on top of the extrusions with neurons faced down to allow a thin film of Neurobasal medium between neurons and glia.

For mass cultures, 6-well plates (Greiner) were coated with sterile $H_2O$ (Baxter) containing 2.5 μg/ml Laminin (Sigma) and 0.0005% Poly-L-Ornithine (Sigma) for at least 2 h at 37°C. Coverslips were then washed three times with sterile $H_2O$ (Baxter), and Neurobasal medium supplemented with 2% B-27, 1.8% HEPES, 0.25% glutamax, and 0.1% Pen/Strep (all Gibco) was added. Cortical neurons were plated at a density of 500K per well.

### SALM1 antibodies

We used a rabbit polyclonal SALM1 antibody targeted against the 638–788 cytoplasmic amino acid fragment of mouse SALM1 developed by Synaptic Systems in all experiments labeling SALM1 except immunoelectron microscopy. For immunoEM, we used a different antibody (ProSci Cat#5067, RRID:AB_10906317) that recognized SALM1 with high specificity (Appendix Fig S1D).

### Plasmids

To test the specificity of the SALM1 antibody, full-length SALM1, SALM2, SALM3, SALM4, and SALM5 were cloned from a yeast two-hybrid cDNA library. A mCherry tag was subcloned between amino acids 20 and 21 of SALM1, and a 3xFlag tag was placed at the N-terminus of SALM2, SALM3, SALM4, and SALM5. All constructs were driven by CMV promoters.

For SALM1 knockdown experiments, two shRNA's directed against SALM1 were acquired from Sigma-Aldrich.
shRNA #1 (shRNA 173892, Sigma-Aldrich):
5′-CCGGGATTCTGGTCATCGGAGGTATCTCGAGATACCTCCGATGACCAGAATCTTTTTTG-3′
shRNA #2 (shRNA 174511, Sigma-Aldrich):
5′-CCGGCAAAGGAAAGAAGAACTTCTACTCGAGTAGAAGTTCTTCTTTCCTTTGTTTTTTG-3′

shRNA-resistant SALM1 and SALM1ΔPDZ were developed against shRNA #2 (rSALM1 and rSALM1ΔZ). Three silent point mutations were made in the SALM1 cDNA to allow translation in the presence of shRNA #2: CAgAGGAAAGAgGAACTTCTg

In the sequence, three small "g" represent the silent mutations. A scrambled shRNA with the following sequence was used as a control:
5′-TTCTCCGAACGTGTCACGT-3′

SALM1 shRNA's and scrambled sequences were subcloned with IRES-GFP to visualize infected cells. SALM1 rescue constructs were subcloned with IRES-mCherry to visualize infected cells. All scrambled, SALM1 shRNA and rescue constructs were driven by synapsin promoters to enhance neuron-specific expression.

For live cell imaging, shRNA#2 and scrambled sequences were subcloned with IRES-ECFP to visualize transfected cells. The constructs were driven by synapsin promoters. Synaptophysin1-pHluorin (SyPhy) was previously characterized (Granseth et al, 2006). The pHluorin tag was subcloned in the intraluminal domain of Synaptophysin1.

To visualize SALM1 (surface) expression in HEK cells and neurons, pHluorin was subcloned between amino acids 526 and 527 of full-length SALM1 (SALM1-pHl). PHluorin-tagged SALM1 lacking its PDZ binding domain (SALM1ΔPDZ-pHl) was generated from the SALM1-pHl construct by PCR. To neutralize charge in the juxtamembrane domain of SALM1-pHl, two point mutations were introduced: R556A and K558A (SALM1$^{RAKA}$-pHl). For expression in HEK cells, expression of all SALM1 constructs was driven by a CMV promoter. For expression in neurons, SALM1-pHl and SALM1$^{RAKA}$-pHl were driven by synapsin promoters.

Neurexin1β(-SS4)-pHluorin (Nxn1β(-SS4)-pHl) was a generous gift from the laboratory of Joris de Wit (VIB Center for the Biology of Disease and Center for Human Genetics, KU Leuven, Belgium) with permission of Z. Josh Huang (Cold Spring Harbor Laboratory, Cold Spring Harbor, NY, USA) and was characterized previously (Fu & Huang, 2010). For the surface expression assays, pHluorin was replaced by a 3xFLAG tag (Nxn1β-FLAG). HA-Neuroligin1AB was a kind gift from Peter Scheiffele (pCAG-NL1AB, Addgene plasmid #15262) (Chih et al, 2006). To be able to directly visualize HA-Nlg1AB in HEK cells for mixed culture assays, HA-Nlg1AB was subcloned with IRES-GFP (HA-Nlg1AB). For expression in HEK cells, all Neurexin and Neuroligin constructs were driven by a CMV promoter. For expression in neurons, Neurexin constructs were driven by synapsin promoters.

PH-PLCδ1-GFP was previously characterized as a marker for PIP2 (Milosevic et al, 2005). To co-visualize the phosphoinositide PIP2 with SALM1-pHl, the GFP tag in PH-PLCδ1-GFP (a kind gift

from I. Milosevic, Departments of Membrane Biophysics and Neurobiology, Max Planck Institute for Biophysical Chemistry, Gottingen, Germany) was exchanged for mCherry (PH-PLCδ1-mCherry).

To investigate the interaction between SALM1 and CASK, full-length CASK was picked up from a cDNA library and was subcloned with IRES-mCherry. The cytoplasmic part of SALM1 was obtained through PCR and was subcloned with a V5 tag at the N-terminus (V5-cytoplasmic SALM1). V5-cytoplasmic SALM1ΔPDZ was obtained by PCR on V5-cytoplasmic SALM1.

### HEK cell transfection

Human Embryonic Kidney 293T cells (HEK cells) were cultured in DMEM/F12 medium with L-glutamine supplemented with 10% FCS, 1% NEAA, and 1% Pen/Strep (all Gibco). Cells were plated in 6-well culture plates (Greiner) at equal densities 1 day prior to transfection at 37°C, 5% $CO_2$. At the day of transfection, cell confluency reached 80%. Cells were transfected using a calcium/phosphate transfection method. Briefly, the desired cDNA at a concentration of 1 μg/μl was diluted 1:40 (or 1:80 in case of co-transfection) in HBS [140 mM NaCl, 1.5 mM $Na_2HPO_4.H_2O$, and 50 mM HEPES adjusted to pH 7.05 with NaOH (all Sigma)]. An equal volume of 250 mM $CaCl_2$ was added dropwise to the DNA-HBS solution under constant mild vortexing. The DNA-HBS-$CaCl_2$ was added dropwise to wells containing HEK cells. Cells were then incubated with the transfection mix for ~20 h at 37°C, 5% $CO_2$. To investigate posttranslational modification of SALM1, 2.5 μg/ml Tunicamycin (Sigma) or 0.25% DMSO (Sigma) was added to the medium after ~3 h of incubation with the transfection mix. Medium was replaced after the ~20-h incubation period, and cells were incubated at 37°C, 5% $CO_2$ for another ~5 h before further use.

### Lentivirus production

To express plasmids in neurons, constructs were cloned into lentiviral backbones and co-transfected with pMD2.G (Addgene, #12259) lentiviral envelope and with pCMVΔR8.2 (Addgene, #12263) Lentiviral packaging into HEK cells. One day after transfection, medium was changed for X10 Opti-MEM(R) (Gibco) supplemented with 100× Pen/Strep (Gibco). Forty hours after transfection, conditioned medium was harvested and spun down at 1,000 $g$ to remove cell debris. Supernatant was transferred to Amicon spin filters (Millipore, 100 kDa cutoff) and spun down at 4,000 $g$ for 30 min. Centrifugation was repeated until all supernatant passed the filter. Concentrated virus was diluted in PBS and filtered with a 0.2 μm filter (VWR). Virus was stored at −80°C until use.

### SALM1 knockdown

To deplete SALM1 from neurons, autaptic neurons were infected at three different time points (DIV2, DIV7, and DIV9) with scrambled, shRNA#1, or shRNA#2 virus. At DIV7, autaptic neurons were also simultaneously infected with shRNA#2 and rSALM1 viruses or with shRNA#2 and rSALM1ΔPDZ viruses. Scrambled, shRNA#1, and shRNA#2 viruses also co-expressed EGFP to control for viral expression. rSALM1 and rSALM1ΔPDZ co-expressed mCherry to control for viral expression. Viruses were titrated such that infection efficiencies reached ~100% for each virus. For all three infection time points (DIV2, DIV7, and DIV9), neurons were analyzed 7 days after infection (DIV9, DIV14, or DIV16).

For sandwich cultures, neurons were infected with scrambled, shRNA#2, shRNA#2+rSALM1, or shRNA#2+rSALM1ΔPDZ viruses at DIV3. Neurons were then analyzed 7 days later at DIV10.

### Western blot analysis

Cultured HEK cells or neurons were lysed with SDS loading buffer and boiled at 90°C for 5 min. Brain tissue was dissociated by shear force in PBS and spun down at 12,000 $g$ at 4°C. PBS was sucked off, and 1× SDS loading buffer was added to the tissue (1 ml loading buffer per 0.1 g tissue). The samples were then boiled at 90°C for 5 min and further homogenized using an insulin syringe (BD MicroFine). Co-IP samples were boiled at 90°C for 5 min and spun down for 1 min at 12,000 $g$. Samples were loaded onto a 1 mm stacking gel consisting of 13.3% Acrylamide/Bis solution, 29:1 (30% w/v) (Serva Electrophoresis), 12.4% 1M Tris (pH 6.8) (AppliChem), 0.2% SDS (VWR Chemicals), 0.1% APS (AppliChem), and 0.01% TEMED (Electran, VWR Chemicals). PageRuler™ Prestained Protein Ladder (Thermo Scientific) was loaded as a reference for molecular weights. Loaded samples were then run through a 8 or 10% running gels consisting of 26.6% (for 8% gels) or 33.2% (for 10% gels) Acrylamide/Bis solution, 29:1 (30% w/v) (Serva Electrophoresis), 24.9% 1.5M Tris (pH 8.8) (AppliChem), 0.13% SDS (VWR Chemicals), 0.07% APS (AppliChem), and 0.007% TEMED (Electran, VWR Chemicals) for 1 h at 50 mA. Proteins were transferred onto PVDF membranes (Immuno-Blot, Bio Rad) at 4°C for 1.5 h at 35V. Blots were blocked with PBS/Tween (PBS supplemented with 0.1% Tween 80 (Ferak Berlin GmbH)) containing 22 mg/ml skim milk powder (Merck) and 4.8 mg/ml Albumin Bovine Serum (Across Organics, Fisher Scientific) for 45 min. at room temperature. Blots were then incubated with primary antibodies diluted in PBS/Tween overnight at room temperature. Blots were washed three times with PBS/Tween and incubated with the following alkaline phosphate secondary antibodies for 1 h at room temperature: alkaline phosphatase AffiniPure goat anti-mouse IgG (H+L) (Jackson ImmunoResearch) and alkaline phosphatase AffiniPure goat anti-rabbit IgG (H+L) (Jackson ImmunoResearch). Blots were washed three times with PBS/Tween and incubated with AttoPhos (Promega) for 5 min. Signals were detected using a FUJIFILM FLA-5000 imaging system and ImageReader FLA5000 software (version 2.0). For quantification of SALM1 levels in cultured neuron lysates, intensity of detected SALM1 bands was determined using the GelAnalyzer tool in ImageJ and was normalized for the intensity of detected actin bands.

### Immunocytochemistry

For surface staining in HEK cells, antibodies were directly diluted in warm DMEM/F12 medium with L-glutamine supplemented with 10% FCS, 1% NEAA, and 1% Pen/Strep (all Gibco). For surface staining of neurons, antibodies were diluted in warm Neurobasal medium supplemented with 2% B-27, 1.8% HEPES, 0.25% glutamax, and 0.1% Pen/Strep (all Gibco). The medium on the coverslips was replaced by primary antibody solution and incubated for 15 min at 37°C, 5% $CO_2$. Cells were washed twice with warm medium and incubated with secondary antibody diluted in warm medium for 15 min. Cells were washed twice with warm medium and fixed with 4% PFA for 30 min at room temperature. After staining, all coverslips were embedded in Mowiol (Calbiochem) or were further processed for intracellular staining (see below).

For intracellular staining, coverslips were fixed using 4% PFA (Electron Microscopy Sciences) for 30 min. Cells were washed three times with PBS and permeabilized for 5 min with 0.5% Triton X-100 (Fisher Chemical). Aspecific binding sites were blocked by incubating cells with 0.1% Triton X-100 (Fisher Chemical) and 2% normal goat serum (Gibco) for 30 min. Cells were incubated for 2 h with primary antibodies. The following primary antibodies were used for intracellular and/or surface staining: rabbit polyclonal SALM1 1:100 (Synaptic Systems), chicken polyclonal MAP2 1:10,000 (Abcam Cat# ab75713, RRID:AB_1310432), guinea pig polyclonal VGluT1 1:5,000 (Millipore Cat# AB5905, RRID:AB_2301751), guinea pig polyclonal Homer 1:300 (Synaptic Systems Cat# 160 004, RRID:AB_10549720), mouse monoclonal Smi312 1:1,000 (BioLegend Cat# 837904, RRID: AB_2566782), guinea pig polyclonal Synaptophysin1 1:250 (Synaptic Systems Cat# 101 004, RRID:AB_1210382), rabbit polyclonal GFP 1:500 (GeneTex Cat# GTX20290, RRID:AB_371415), guinea pig polyclonal GFP (Synaptic Systems Cat# 132 005, RRID:AB_11042617), mouse monoclonal GFP (Thermo Fisher Scientific Cat# 14-6674-82, RRID:AB_2572900), mouse monoclonal mCherry 1:1,000 (Signalway Cat# T515, RRID:AB_2721246), rabbit polyclonal mCherry (GeneTex Cat# GTX128508, RRID:AB_2721247), mouse monoclonal Flag 1:1,000 (Sigma-Aldrich Cat# F1804, RRID:AB_262044), and/or mouse monoclonal HA 1:500 (Roche Cat# 11666606001, RRID:AB_514506). After primary antibody incubation, cells were washed three times with PBS and incubated with secondary antibodies for 1 h. The following secondary antibodies were used for intracellular and/or surface staining: goat anti-mouse Alexa 488, 546, and 647, goat anti-rabbit Alexa 488, 546, and 647, goat anti-guinea pig Alexa 488, 546, and 647, and goat anti-chicken Alexa 647 (all Invitrogen). To stain for F-actin, Phalloidin-Rhodamine 1:10 (Thermo Fisher Scientific Cat# R415, RRID:AB_2572408) or Phalloidin-Alexa 647 1:10 (Thermo Fisher Scientific Cat# A22287, RRID:AB_2620155) were added to the secondary antibody mix. Coverslips were then washed three times with PBS and embedded in Mowiol (Calbiochem). All steps were performed at room temperature.

### Confocal image acquisition and analysis

Stained cultures were imaged with a Zeiss LSM510 confocal microscope equipped with a 63× oil objective (N.A 1.4, plan apochromat) or a 40× oil objective (N.A 1.3, plan neoflux). Images were acquired using an AxioCam MRm (Zeiss) camera and Zeiss LSM510 software (version 4.2). All images were acquired at room temperature. For mixed cultures, Z-stack images were acquired with set intervals (0.3 μm). Gain and amplifier offset were kept constant for different conditions for each experiment. For neuronal cultures, neurite morphology, synapse numbers, and synapse intensities were determined using SynD semi-automated analysis software (Schmitz *et al*, 2011). The GFP signal expressed by neurons infected with scrambled or SALM1 shRNA's was used in the morphology channel to determine cell morphology. SynD software was also used to quantify Neurexin and SALM1 clusters in neuronal cultures overexpressing SALM1-pHl and/or Nxn1β(-SS4)-FLAG. In these experiments, actin was used as morphology marker. ImageJ (version 1.50a) was used to analyze mixed cultures. Briefly, z-stack images were collapsed using the Z Project Max Intensity option. Channels were split, and background subtraction using a rolling ball algorithm was applied to the channel showing the synapse marker. A mask was drawn around the HEK cell (GFP and pHluorin tags were used to show HEK cell

morphology), and a threshold was set such that only synapses within the HEK cell mask were detected by the Analyze Particles tool. The threshold was kept constant amongst different conditions for each experiment. Experiments of Neurexin and SALM1 expression in HEK cells were also analyzed using ImageJ software. Briefly, single z-slice images were transformed using the Polar Transformer plugin. A mask was then drawn around the cell membrane. Neurexin clusters were selected using the Niblack auto local threshold option with parameter1 = 2 and parameter2 = 0 followed by the analyze particle tool. Average cluster intensity was determined using the acquired mask. Average intensity outside clusters was determined by combining the membrane mask and the cluster mask using the XOR option in the ROI Manager.

### Electron microscopy

For immunogold labeling of SALM1, hippocampi of P75 mice were fixed using 4% PFA (Electron Microscopy Sciences) and 0.1% glutaraldehyde (Merck) in 0.1 M phosphate buffer (pH 7.4). Samples were embedded in increasing concentrations of gelatin at 37°C, infiltrated with 2.3 M sucrose at 4°C, and frozen in liquid nitrogen. A cryo-ultramicrotome (UC6, Leica) was used to cut tissue in 70-nm-thick sections. These were collected in 1% methyl-cellulose and 1.2 M sucrose solution at −120°C. The sections were then transferred to formvar/carvon-coated copper mesh grids, washed with PBS at 37°C, and treated with 0.1% glycine. Hippocampal sections were immunolabeled using a primary antibody against SALM1 (ProSci Cat#5067, RRID:AB_10906317) diluted 1:10 in PBS supplemented with 0.1% BSA. Antibodies were further labeled using Protein A-10 nm gold (CMC, UMC Utrecht, Netherlands). Sections were counterstained on ice with 0.4% uranyl acetate in 1.8% methyl-cellulose and were imaged on a Tecnai 12 BioTwin transmission electron microscope (FEI Company).

For ultrastructural analysis of SALM1-depleted synapses, mouse autaptic hippocampal neurons were infected at DIV7 with scrambled, shRNA#1, or shRNA#2 and were fixed at DIV14 with 2.5% glutaraldehyde (Merck) in 0.1M cacodylate buffer (pH 7.4) for 1 h. Cells were washed and stained for 1 h with 1% $OsO_4$/1% $KRu(CN)_6$ in milliQ water at room temperature. Samples were then embedded in epoxy resin and cut into 80-nm-thick sections. Cell sections were further stained using uranyl acetate and lead citrate in Ultra stainer LEICA EM AC20 and were imaged with a JEOL1010 electron microscope (JEOL, Tokyo, Japan). Images were acquired using a side-mounted CCD camera (Morada; Olympus Soft Imaging Solutions, Münster, Germany) and iTEM analysis software (Olympus Soft Imaging Solutions). Imaging and analysis was performed blinded.

### Electrophysiology

Hippocampal autaptic whole-cell voltage-clamp electrophysiology was performed at room temperature (20–23°C) at DIV14-15 (for neuron infected at DIV 7) and at DIV16-17 (for neurons infected at DIV 9). Cells were kept in a voltage clamp (membrane potential, $V_m = -70$ mV) using Axopatch 200B amplifier (Axon Instruments) with borosilicate glass pipettes (2–4 MΩ) containing 125 mM $K^+$-gluconic acid, 10 mM NaCl, 4.6 mM $MgCl_2$, 4 mM K2-ATP, 15 mM creatine phosphate (Calbiochem), 1 mM EGTA, and 20 U/ml phosphocreatine kinase (pH 7.3, 300 mOsm). All chemicals were from Sigma-Aldrich unless stated otherwise. Electrical stimulation was performed by depolarization from −70 to 30 mV for 0.5 ms. Series

resistance was always 70% compensated, and cells with holding current lower than −300 pA or with resistance above 10 MΩ were discarded. The external solution contained 140 mM NaCl, 2.4 mM KCl, 4 mM $CaCl_2$, 4 mM $MgCl_2$, 10 mM HEPES, and 10 mM glucose (pH 7.3, 300 mOsm). Paired-pulse protocol was applied at different frequencies: 20, 50, 100, 200, 500, and 1,000 Hz. Train stimulations were given of 60 stimulations at 10 Hz. Digidata 1440A and pCLAMP 10 software (Axon instruments) were used to record all signals.

### Live cell imaging

Wild-type neurons on continental cultures were transfected with calcium phosphate as described previously (Köhrmann et al, 1999) with SypHy in combination with ECFP-tagged scrambled or shRNA#2 at DIV 7. Coverslips were placed in an imaging chamber and perfused with Tyrode's solution (pH7.4) consisting of 119 mM NaCl, 2.5 mM KCl, 2 mM $CaCl_2$, 2 mM $MgCl_2$, 25 mM HEPES, and 30 mM Glucose (VWR) (all Sigma unless stated otherwise). 5 μM AP5 (Ascent) and 10 μM DNQX (Tocris) were added to the Tyrode's solution to block NMDA and AMPA receptors, respectively. Transfected neurons were imaged on a Zeiss Axiovert 200 M microscope equipped with a CoolSNAP camera HQ (Photometrics) and an illumination unit (Polychrome IV). Images were acquired with MetaMorph software (version 6.2r6; Universal Imaging) at 1 Hz using a 40× oil objective, NA 1.3. Electrical field stimulation was applied through parallel platinum electrodes by an IsoFlex Stimulus Isolator (A385; World Precision Instrument), regulated by a Master-8 pulse generator (A.M.P.I.), providing 300 action potentials (AP) at 10 Hz with an output of 30 mA. Imaging protocol consisted of 15-s image acquisition at 1 Hz, 30 s of electrical stimulation followed by a recovery period of 70 s. The imaging protocol ended with 5-s superfusion of ammonium chloride Tyrode's buffer consisting of 69 mM NaCl, 50 mM $NH_4Cl$, 2.5 mM KCl, 2 mM $CaCl_2$, 2 mM $MgCl_2$, 25 mM HEPES, and 30 mM Glucose (VWR) (all Sigma unless stated otherwise) followed by a 10-s recovery period to visualize total pool size after stimulation. All imaging experiments were performed at RT.

Stacks from time-lapse recordings acquired with 1-s interval were used to analyze sypHy fluorescence. Synaptic vesicle release events (characterized as puncta with a gradual increase in fluorescence) were detected by visual inspection in ImageJ. A $6 \times 6$ pixel (corresponding to $2.4 \times 2.4$ μm) region of interest (ROI) was centered on the event, and average intensity of fluorescence was measured. Moving vesicles and silent synapses (not responding to the stimulus) were excluded from analysis. Fluorescent traces were plotted as change in fluorescence ($\Delta F$) normalized to the initial fluorescence ($F_0$), obtained by averaging the 15 frames before onset of stimulus, as a function of time. The maximal change in fluorescence due to the electric stimuli (300AP) was called F300. $F_{max}$ was defined as the average of the 5 highest values during $NH_4Cl$ Tyrode's solution superfusion of each ROI. Onset of exocytosis was defined as the first frame in which the average fluorescence increased above a threshold of $\Delta F/F_0 = 0.1$.

### Mixed culture assays

Co-cultures of heterologous HEK cells and neurons were performed according to Biederer and Scheiffele (2007). Briefly, HEK cells were transfected using calcium phosphate transfection with cDNA plasmids expressing either GFP, SALM1-pHl, Nxn1β(-SS4)-pHl, or

HA-Nlg1AB. One day after transfection, HEK cells were collected and dissociated by pipetting medium up and down several times. Cells were counted and seeded at a density of 15k per coverslip on top of wild-type DIV9 mouse hippocampal neurons cultured in a sandwich culture system. For investigating the effect of SALM1 depletion on Neurexin- and Neuroligin-mediated synapse formation, HEK cells expressing Nxn1β(-SS4)-pHl or HA-Nlg1AB were plated at a density of 15K on top of DIV9 mouse hippocampal neurons that were infected at DIV3 with scrambled, shRNA#2, shRNA#2+r-SALM1, or shRNA#2+rSALM1ΔPDZ. One day after seeding, co-cultures were fixed and further examined using immunocytochemistry.

### Surface expression assays

To investigate surface expression patterns of individual cell adhesion molecules, HEK cells were transfected with SALM1-pHl, SALM1ΔPDZ-pHl, Nxn1β(-SS4)-Flag, or HA-Nlg1AB-IRES-GFP alone. To investigate if SALM1 influences the surface expression of Neurexin and Neuroligin, HEK cells were co-transfected with SALM1-pHl and HA-Nlg1AB, with SALM1-pHl and Nxn1β(-SS4)-Flag, or with SALM1ΔPDZ-pHl and Nxn1β(-SS4)-Flag. One day after transfection, HEK cells were collected and dissociated by pipetting medium up and down several times. Cells were counted and seeded at a density of 15k per coverslip on glass coverslips coated with 2.5 μg/ml Laminin (Sigma) and 0.0005% Poly-L-Ornithine (Sigma). One day after seeding, surface staining was performed as described under immunocytochemistry.

### Co-immunoprecipitation

To investigate potential interactions between SALM1 and Neurexin, HEK cells were co-transfected with SALM1-pHluorin and/or Nxn1β (-SS4)-FLAG. To investigate the potential interaction between SALM1 and CASK, HEK cells were transfected with full-length CASK alone or together with V5-tagged cytoplasmic SALM1, V5-tagged cytoplasmic SALM1ΔPDZ, or empty vector V5-pcDNA3.1. One day after transfection, medium was replaced for DMEM/F12 medium with L-glutamine supplemented with 10% FCS, 1% NEAA, and 1% Pen/Strep (all Gibco). The next day, cells were scraped and collected in lysis buffer containing 50 mM Tris (pH7.5) (AppliChem), 1% Triton X-100 (Fisher Chemicals), 1.5 mM $MgCl_2$ (Sigma), 5 mM EDTA (Applichem), 100 mM NaCl (Sigma), and 1× Protease inhibitor (Sigma) on ice. Lysates were spun down for 10 min. at 12,000 $g$ at 4°C. Part of the supernatant was mixed with 5× SDS loading buffer and stored at −20°C to use later as input control. Protein A agarose beads were washed once with lysis buffer and combined with the remaining lysate. Samples were then tumbled for 1 h at 4°C followed by 1 min. centrifugation at 12,000 $g$. Supernatant was transferred to new tubes and incubated for 2 h at 4°C with the following antibodies: rabbit polyclonal GFP (0.3 μl/IP) (GeneTex Cat# GTX20290, RRID:AB_371415), mouse monoclonal FLAG (0.3 μl/IP) (Sigma-Aldrich Cat# F1804, RRID:AB_262044), mouse monoclonal V5 (Abcam Cat# ab27671, RRID:AB_471093), and mouse monoclonal CASK (UC Davvis/NIH NeuroMab Facility Cat#73-000, RRID:AB_10671954). Protein A agarose beads were pre-blocked with blocking buffer existing of lysis buffer supplemented with 20% glycerol (VWR, BDH) and 0.2% chicken egg albumin (Sigma) for 1 h at 4°C. Beads were then washed once with lysis buffer and mixed with the lysate samples. After 1 h of incubation at

4°C, beads were washed five times alternately with low salt buffer consisting of 50 mM Tris (pH7.5) (AppliChem), 0.1% Triton X-100 (Fisher Chemicals), 1.5 mM $MgCl_2$ (Sigma), 5 mM EDTA (AppliChem), 100 mM NaCl (Sigma), and 1× Protease inhibitor (Sigma) and with high salt buffer containing 50 mM Tris (pH7.5) (AppliChem), 0.1% Triton X-100 (Fisher Chemicals), 1.5 mM $MgCl_2$ (Sigma), 5 mM EDTA (AppliChem), 200 mM NaCl (Sigma), and 1× Protease inhibitor (Sigma). Wash buffer was then fully removed using an insulin syringe (BD MicroFine), and SDS loading buffer was added to the beads. Samples were stored at −20°C until further use.

**Quantification and statistical analysis**

*Statistics*

Statistical analysis was performed using SPSS (IBM SPSS Statistics 21). Kolmogorov–Smirnov and Shapiro–Wilk tests were used to test for normality of data distribution. A one-way ANOVA with post hoc Bonferroni test (or Games-Howell when Levene's test was significant) was used to compare three or more groups when data were normally distributed. A Kruskal–Wallis test with post hoc pairwise comparisons was applied for comparison of three or more groups when data were not normally distributed. A Mann–Whitney *U*-test was used to compare two groups when data were not normally distributed. Bonferroni correction was used in case of multiple Mann–Whitney *U*-tests within one data set.

Expanded View for this article is available online.

## Acknowledgements

We thank Ingrid Saarloos, Robbert Zalm, and Joost Hoetjes for cloning, producing viral particles, and providing HEK cell cultures; Desiree Schut and Frank den Oudsten for providing primary neuron cultures; Joke Wortel for animal breeding; Rien Dekker for EM microscopy and analysis (the EM facility is supported by ZonMW 91111009); Jurjen Broeke for technical assistance with confocal microscopy; we thank all members of the Verhage laboratory for helpful discussions. We thank Synaptic Systems for providing SALM1 antibody. We thank Joris de Wit and Josh Huang for generously providing tagged Neurexin1β plasmid. This work is supported by an ERC Advanced Grant (322966) of the European Union (to MV).

## Author contributions

Conceptualization, MB, FF, RFT, and MV; Investigation, MB, FF, FK, NC, SO and JWB; Writing/Visualization, MB, RFT, and MV; Supervision, KWL, JRTW, ABS, RFT, and MV; Funding acquisition, MV.

## Conflict of interest

The authors declare that they have no conflict of interest.

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
