## [Review Process File · The EMBO Journal]

SALM1 controls synapse development by promoting F-actin/PIP2 dependent Neurexin clustering

Marinka Brouwer, Fatima Farzana, Frank Koopmans, Ning Chen, Jessie W. Brunner, Silvia Oldani, Ka Wan Li, Jan R. T. van Weering, August B. Smit, Ruud F. Toonen & Matthijs Verhage

Review timeline:

Submission date:	3rd Dec 2018
Editorial Decision:	14th Jan 2019
Revision received:	13th May 2019
Editorial Decision:	6th Jun 2019
Revision received:	20th Jun 2019
Accepted:	25th Jun 2019

Editor: Karin Dumstrei

Transaction Report:

1st Editorial Decision

14th Jan 2019

Thank you for submitting your manuscript to The EMBO Journal. Your submission has now been reviewed by three good experts in the field and their comments are provided below.

As you can see from the comments below, the referees find the analysis interesting and suitable for publication here. They raise a number of relevant concerns that are clearly outlined below. Given the comments raised, I would like to invite a suitably revised manuscript that addresses the concerns raised.

REFeree REPORTS:

Referee #1:

The spatio-temporal regulation of synaptogenic mechanisms and the cooperation of synapse-organizing molecules are barely understood. Brouwer and colleagues present in this study an analysis of the adhesion molecule SALM1. They identify it in an interaction screen as a membrane protein binding to the PDZ domain proteins CASK and Lin7b. Immunocytochemistry shows that SALM1 is enriched at excitatory synapses and immunoEM localizes it to pre- and postsynaptic specializations. Co-culture assays of neurons in which SALM 1 was knocked-down using lentivirus demonstrate that both Neurexin1beta- and Neuroligin1-mediated formation of post- and pre-synaptic assemblies is impaired when SALM1 expression is reduced. Further, SALM1 acts in a PDZ interaction-dependent manner to promote the synaptogenic activity of these two proteins. The authors then analyze cell biological effects of SALM 1 on Neurexin1beta and Neuroligin-1 expression in heterologous cells. This shows that SALM1 clusters at plasma membrane domains enriched for F-actin and PIP2 and assembles proximal to Neurexin1beta. This property is shared by all SALM family members and requires a polybasic intracellular region similar to charged stretches

in other proteins binding PIP2. The polybasic region-dependent Neurexin clustering by SALM1 is also observed in neurons. Consistent with a role in synapse development, knockdown of SALM1 in neurons prior to 8 div decreases synaptic vesicle recruitment as shown by immunocytochemistry and EM, and reduces EPCS amplitude and mini frequency at 14/15 div.

This study provides interesting insights into a SALM1- and PIP2-dependent mechanism that locally clusters synaptogenic Neurexin molecules and controls their activity. This also reveals an unexpected interplay of these two synaptic membrane proteins. Points to further strengthen this work are provided below.

Major point.

1. This study reports two key findings. First, Neurexin1beta clustering can be induced by SALM1, and second, SALM1 is required for the recruitment and recycling of excitatory synaptic vesicles and supports excitatory transmission. The title states that "SALM1 controls synapse development by promoting F-actin/PIP2 dependent Neurexin clustering". Yet, a causal relationship of the two findings is not shown. Do the authors have evidence that presynaptic functions of SALM1 requires Neurexin1beta clustering? At a minimum, they should analyze whether the SALM1 PIP2-binding RAKA mutant rescues vGlut1 clustering in SALM1 knockdown neurons (Fig. 5C,F) and whether the mutant rescues in the pHluorin assay (Fig. 6). Both experiments are relevant for our understanding of roles of SALM1 in PIP2/F-actin dependent presynaptic assembly.
2. Does SALM1 knock-down in neurons disperse Neurexin1beta at presynaptic, vGlut1-positive sites along neurites and can this be rescued by the SALM1 RAKA mutant (or not)? This would support a requirement of SALM1 for synaptic retention or clustering of Neurexins.

Minor points.

3. Does the SALM1-induced clustering of Neurexin1beta in turn cluster Neuroligin-1 at cell-cell contact sites? This would be informative to assess roles of SALM1 in trans-synaptic assembly.
4. The use of autapses precludes assessment of whether SALM1 functions pre- or post-synaptically as it is knocked-down on both sides. Fig. 5A-G is a good example for this issue. The images show dendrites with fewer and less intense vGlut1 puncta upon SALM1 knockdown but it is not clear whether this involves presynaptic loss of SALM1. Do the authors have experiments beyond the co-cultures that allow to address the site of SALM1 action for the effects analyzed here?
5. It would be quite informative if the authors have results how SALM1 promotes Neurexin1beta clustering at F-actin/PIP2 microdomains.
6. SALM1 knockdown from 9 div does not reduce synaptic transmission, unlike the knockdown starting at 7 div. This is a rather precisely defined window of requirement; can the authors discuss what developmental maturation steps may cause this loss of SALM1 requirement?
7. Why do the other SALMs not compensate for knock-down of SALM1, is there information about their expression profile in developing neurons?
8. The abstract states that SALM1 is preferentially presynaptic. Based on the EM quantification, there is no striking enrichment of SALM1 in pre- vs post-synaptic compartments, though, and if one only looks at synaptic membranes, it is actually twice as abundant postsynaptic. The statement in the abstract should be rephrased and this info about pre/postsynaptic membrane abundance can be stated in the text of the Results.
9. Figure 5B/C/D refers to 'synapse number' as counted parameter, but rather shows vGlut1 puncta abundance. The legend can be updated. Also, how was puncta abundance measured, are these puncta per ROI and how was the ROI defined? Why was not a standard measure obtained, e.g. puncta number per dendrite length?
10. The IP screen did not identify known CASK binding partners like Neurexins, SynCAMs, or

Syndecans. This can be stated to communicate that this approach yielded interesting new partners but does not cover them all, as can be expected.

11. The Figure Legends can provide the approaches for the overexpression and knock-down in neurons, i.e. lenti vs calcium phosphate. This helps readers to understand the experiments.

13. The Methods section refers to multiple different types of neuronal cultures used in this study, including autapses. The info which types of culture were analyzed is not provided in the Results and Figure Legends, making assessment of the data a bit of educated guess work. This needs to be added.

14. There is a typo in Figure 4B, Phluorin instead of pHluorin.

Referee #2:

In this study "Adhesion molecule SALM1 controls synapse development by promoting F-actin/PIP2 dependent Neurexin clustering" by Brouwer et al., the authors interrogate the presynaptic function of SALM1. Unlike SALM2-5, which are localized postsynaptically, SALM1 is located to pre- and post-synaptic terminals. The authors demonstrate that SALM1 directly interacts with the presynaptic organizing protein, CASK and, in HEK293 cells can promote the clustering of Nrnx1 β , via indirect cis-interactions, in PIP2 enriched microdomains. These initial findings position SALM1 to be a critical node that mediates the organization of the presynaptic terminal. In HEK293 cells and neurons, the ability of SALM1 to cluster Nrnx1 β is independent of its PDZ binding motif, but is dependent on a SALM1 polybasic region. The authors show that knockdown of endogenous SALM1 reduces excitatory synapse number, reduces the size of the vesicle pool without a change in the RRP and impairs excitatory synaptic transmission. The authors attribute impaired synaptic transmission to reduced vesicular fusion.

I am particularly fascinated by the novel mechanism proposed by the authors that clusters Nrnx1b in HEK293 cells because it has been assumed that postsynaptic ligands (e.g. neuroligins) promote the clustering of presynaptic neurexins, however, the data presented here would suggest that presynaptic SALM1 promotes clustering of neurexins BEFORE their interactions with neuroligins. This raises the intriguing possibility that neurexins are already primed to be central organizers of the presynapse and only need to engage in transsynaptic interactions to activate these signaling pathways. While these findings are the first to characterize the presynaptic role of SALM1, identify a potential mechanism for cis-interactions that cluster neurexins and have the potential to be of great interest, my enthusiasm is tempered because the link between SALM1-dependent Nrnx1 β clustering and synapse formation and synaptic transmission is tenuous - the critical conditions/controls to test whether Nrnx1 β clustering via SALM1 in PIP2 microdomains is required for synaptic transmission and synapse formation were not performed.

Major Concerns:

1) It is difficult to easily interpret the results because type of cultures used is not clearly stated in the text, figures or figure legends. Also, the experimental approaches in many of the figures and in the text are lacking and only available in the figure legend (e.g. surface staining vs total). The authors need to ensure that all panels are referred to in the text (e.g. 1D (SMI staining) and 3H).

2) While a significant number of figures are devoted to suggest that SALM1 clusters Nrnx1 β on the cell surface via a polybasic motif in PIP2 microdomains that is PDZ-binding motif independent, the actual functional consequence of this clustering appears to be negligible. In neurons, the PDZ-binding motif of SALM1 is necessary for synapse formation and synaptic transmission. If the polybasic mutations and RAKA+ Δ PDZ mutations are introduced into these assays (unclustered neurexins), how are synapse formation and synaptic transmission impacted? Thus SALM1 appears to intriguingly have PDZ-dependent and PDZ-independent functions and suggest that neurexin clustering is not the primary functional role of presynaptic SALM1.

3) The figures of this paper essentially tell two distinct stories (see point 2) and both parts of the story (PDZ-dependent synapse function and PDZ-independent Nrnx1 β clustering) and experiments/conditions to link the two halves should be included in the manuscript.

4) Antibody: Validate the antibody by using shRNA2 in WB. Do low molecular weight bands disappear?

Are the low MW bands membrane proteins? Is that why the shRNA2 ICC still has significant

SALM1 surface staining?

5) While the authors show that co-transfection of SALM1 without its PDZ-binding motif and Nrnx1 β results in neurexin clustering (Fig. S3L), the authors should use this condition in the artificial synapse formation assay along with proper controls to test if clustering of Nrnx1 β on HE293 cells does anything in this assay. Does the presence of SALM1 (or SALM2-5) somehow enhance the number/size of Homer puncta made onto HEK293 cells?

6) The functional role of Nrnx1 clustering at PIP2 microdomains is unclear. The authors only test SALM1 Δ PDZ's ability to rescue the shRNA-mediated evoked EPSC phenotype, however, the RAKA mutant should also be tested because, while the polybasic motif appears to be necessary to facilitate Nrnx1 β clustering, the potential contribution of this motif to morphology and function was not tested.

7) The authors' conclusion that the striking reduction in basal synaptic transmission is in part due to impaired vesicular release following SALM1 depletion is questionable. First, in Fig 5H, EM shows no change in docked vesicles following SALM1 shRNA, but a decrease in total vesicle numbers. Single action potential evoked EPSCs are commonly thought to reflect vesicles released from the docked pool (RRP) and, to an extent, vesicles involved in mEPSCs are also provided by the RRP. Second, when normalized to the deacidified condition, the SypHy-PH1 experiment showed no differences. Because the shRNA changes the number of presynaptic vesicles/bouton (Fig. 5), it is misleading to directly compare the absolute F300 between scrambled and shRNA. The mobilization of the recycling and reserve pool by 10Hz stimulation will obviously be dependent on the total number of vesicles/bouton. Third, electrophysiological measurements during 10Hz stimulation revealed no changes in the normalized EPSC amplitude and apparently no changes in decay kinetics, indicating that vesicle release is intact. Fourth, late infection of SALM1 shRNA does not alter synapse numbers and does not alter synaptic transmission. Because vesicular fusion is fine-tuned and is highly dynamic, one would assume that depletion of SALM1 at any developmental timepoint would perturb this system, independent of changes in synapse density. Together, it is likely that these changes in basal synaptic transmission can simply be explained by reduced synapse numbers observed in Fig. 4.

Minor Concerns:

Figure 1

Figure 1D:

What kind of preparation is this? Was SALM1 live surface stained?

Were the inset vGluT1 image from the representative neuron or acquired from a different neuron?

Was the Homer staining was done in separate neuron preps and, if so, there should be a representative neuron image similar to vGluT1.

Timecourse of SALM1 staining (Fig 1D) to match the synapse morphology experiments done later in the paper to confirm that SALM1 is expressed presynaptically.

The SMI staining is not convincing that endogenous SALM1 is axonal. There is very little/no overlap between the red and green channels. The meaning of SMI staining is not mentioned in the text.

Figure 1G:

The Immuno EM does not show SALM1 particles near the active zone (as compared to neurexin/neurologins. See Burch et al., PLOS ONE. 2017).

Figure 2:

In the next figure, the authors point out that SALM1 causes the clustering of Nrnx1 β in HEK293 cells. The authors should quantify artificial synapses assay comparing HEK293s expressing Nrnx1 β alone vs SALM1 and Nrnx1 β . Does Nrnx1 β enhance hemi-synapse formation?

Figure 3:

Destabilizing F-actin has a dramatic impact on Nrnx1 β clustering, if F-actin polymerization is enhanced, does one see an opposite effect? Does PIP2 depletion prevent Nrnx1 β clustering by SALM1?

3H is not mentioned in the text and is also found in the supplemental figures.

Is CASK expressed in HEK cells? If so, would shRNA KD impair Nrnx1 β 's clustering in the presence of SALM1?

Figure S3:

Overexpression of SALM1 appears to increase the size of PIP2 microdomains in HEK293 cells.

Does SALM1 drive the formation of PIP2 microdomains in a PDZ binding motif independent

manner that traps Nrnx1 β ? The SALM1 Δ PDZ experiments in S3L suggest that close proximity of SALM1 with Nrnx1 β is not necessary to mediate Nrnx1 β clustering.
Is CASK expressed in HEK cells? If so, would shRNA KD impair Nrnx1 β 's clustering in the presence of SALM1?

How microdomains (PIP2) are defined should be explained around line 168-169 instead of on line 190.

Figure 4:

Does the enrichment of SALM1 and Nrnx1 β at PIP2 microdomains in HEK293 cells also occur in neurons? The authors only stain for total actin.

4A: It is curious that FLAG-Nrnx1 β alone does not traffic to the surface of neurons. Others have seen robust trafficking (Savas, JN. Neuron 2015) and expression of Nrnx1 β can functionally rescue synaptic transmission (Aoto, J. Nature Neuroscience 2015), arguing that these molecules traffic to the plasma membrane/active zone.

Does the ratio of surface/intracellular Nrnx1 β increase with SALM1 compared to no SALM1 and SALM1 RAKA?

Live surface labeling artificially clusters surface proteins. It is surprising that there are fewer Nrnx1 β puncta in the RAKA condition, indicating that less Nrnx1 β made it to the surface of these neurites. Does overexpression of SALM1RAKA block Nrnx1 β surface trafficking? To determine this, it would be important to measure total surface and/or surface recycling of Nrnx1 β -FLAG levels for SALM and SALM1RAKA overexpression conditions.

Figure S4B:

SALM2-5 have a polybasic motif and are surprisingly not clustered but rather diffuse. However, these molecules still have the ability to cluster Nrnx1 β . This is confusing and does not support the hypothesis that F-actin/PIP2 trap SALM oligomers, which then clusters Nrnx1 β . Can the authors explain this observation?

PIP2/F-actin should be imaged with SALM2-5.

Figure 5:

SALM knockdown by shRNA1 is ~50% while shRNA2 is ~80%, yet both display similar loss of synapses. Rescue should be performed for shRNA1 to exclude off-target effects.

It is unclear why the endpoint for staining changes depending on the time of lentiviral transduction because synapse density and morphology (e.g. on DIV 9, all synapses are on shafts on DIV14 and 16, most excitatory synapses are located on spines).

The authors should show merged ICC images. Also, does presynaptic localization of CASK change in the SALM1 shRNA or SALM1 shRNA + rSALM1 Δ PDZ conditions?

Fig. 5B: rescue experiments should be performed.

Fig. 5B, C D: A description of how the analysis was done is needed as it appears that the total number of synapses in an image were quantified (Synapse numbers are usually normalized to unit length because absolute numbers of synapses are dependent on density, which varies from ROI).

Figure 6:

The readily releasable pool is responsible for the vesicles used for basal synaptic transmission (single evoked synaptic transmission Fig. 6A and miniature transmission Fig. 6C). Typically, the recycling pool and/or reserve pool participate in transmission following prolonged repeated stimuli. In Fig. 5M, the number of docked vesicles (RRP) is unchanged. The observed change in total vesicle numbers (Fig. 5I) is likely a change in recycling vesicles and/or reserve pool, however, the electrophysiological methods applied in Figure 6A-F do not test the contribution of the recycling pool. It is thus unclear how the change in total vesicle numbers without a change in docked vesicles can contribute to the synaptic phenotype following SALM1 KD.

The significance of SALM1 mediated clustering of beta neurexin-1 has not been tested. Instead, the entire functional phenotype is dependent on the PDZ binding motif of SALM1. Rescue experiments with RAKA mutant for evoked EPSC and phluorin experiments should be performed.

Can SALM2-5 rescue the SALM1 KD phenotype? When overexpressed, are these family members redundant?

It is confusing why infection of shRNA1 at DIV7 significantly reduces synapse numbers at DIV 14, but fails to manifest as a change in mEPSC frequency but does impair evoked EPSCs.

The RRP should be measured.

Are presynaptic calcium dynamics altered by SALM1 KD?

The decay tau of the train stimuli should be quantified and the train duration should match the phluorin experiment to test if impaired vesicle fusion is observed electrophysiologically. Interpretation of Phluorin experiments should be performed on the data normalized to NH₄Cl. It is not accurate to compare absolute ΔF values. The authors should perform additional experiments to test their hypothesis that vesicle fusion is impaired in by SALM1 shRNA because the functional electrophysiological data argue that vesicle fusion is not the primary cause of the phenotype.

Referee #3:

- General summary and opinion:

Brouwer and colleague investigated the role of the adhesion protein SALM1 in synaptogenesis in vitro. Although they show that SALM1 in itself is not sufficient in aggregating synaptic vesicles, it acts to cluster Neurexin at the presynaptic terminal, leading to the cluster of presynaptic vesicles. I think the study is very well executed and the demonstration convincing. Nevertheless, I have concerns regarding the reporting of the statistics, which does not allow the validation of the data. I would recommend the publication only after clarification of the statistics. This said, I am very enthusiastic about the results and the quality of the demonstration.

- Specific major concerns essential to be addressed to support the conclusions:

1- The statistical tests that the authors used for each of their quantification is not specified. There is only a general mention in the method section, which is not sufficient. It is important to know if they can perform the multiple comparisons.

2- The mention of the number of cells and the number of culture helpful but it is not clear which N is used for quantification (An average per cell or an average per culture?). In figure 5 for the quantification of the EM, I could not work out what the numbers in the bar refer to. Ideally the average should be per culture to avoid problems associated with artificial increase of the power of the statistical tests.

3- I may have missed it but I did not see how many times the western blot in figure 1 (IP) was replicated.

4- I think the authors should include a negative control for their antibody in Figure 1. They have a siRNA that efficiently decrease the expression of SALM1 using western blot and it would be important to see the effect of the siRNA on the immunostaining.

- minor concerns that should be addressed.

1- In their experiments in figure2 they authors conclude that knocking down SALM1 interfere with Neurexin and Neuroligin function leading to a decrease in pre or postsynaptic clustering. An alternative explanation is that SALM1 knockdown would decrease Neuroligin or Neurexin expression levels leading to a decrease in clustering. The authors should control for Neurexin and Neuroligin expression levels in SALM1 KD cultures.

2- The authors do not explain why they use a SALM-pHl construct. I presume that the overexpression lead to a massive expression of SALM1 in vesicles interfering with the visualisation of SALM1 at the membrane. The use of pHl would quench SALM1 expressed in the vesicles and allow the visualisation of membrane SALM1 only. If this is the case of if the reason is different the authors should be explicit about it. They should also comment on the level of overexpression and how it could interfere with their results.

1st Revision - authors' response

13th May 2019

REPLY TO THE REVIEWERS manuscript # EMBOJ-2018-101289

We are grateful to all three reviewers for their insightful comments and strong support for our manuscript and we are honored by their positive qualifications of the work: “*interesting insights*” (reviewer 1); I am very enthusiastic about the results and the quality of the demonstration (reviewer 3). We have revised the manuscript in accordance with their comments, added 13 new data sets (Figures 2, 6, 7, 8, EV2, EV3, S2, S3, S5) to fully comply with their requests. Furthermore, we added better analyses, better descriptions of experiments and better discussion of the results. These changes have greatly strengthened the paper and it’s main message. Please find below a point-by-point response to the reviewers with their original text in black and our response in blue.

Referee #1: This reviewer raises 2 major and 11 minor issues.

“This study provides interesting insights into a SALM1- and PIP2-dependent mechanism that locally clusters synaptogenic Neurexin molecules and controls their activity”.

We thank the reviewer for his/her positive comments and suggestions.

Major points:

1. This study reports two key findings. First, Neurexin1beta clustering can be induced by SALM1, and second, SALM1 is required for the recruitment and recycling of excitatory synaptic vesicles and supports excitatory transmission. The title states that "SALM1 controls synapse development by promoting F-actin/PIP2 dependent Neurexin clustering". Yet, a causal relationship of the two findings is not shown. Do the authors have evidence that presynaptic functions of SALM1 requires Neurexin1beta clustering? At a minimum, they should analyze whether the SALM1 PIP2-binding RAKA mutant rescues vGluT1 clustering in SALM1 knockdown neurons (Fig. 5C,F) and whether the mutant rescues in the pHluorin assay (Fig. 6). Both experiments are relevant for our understanding of roles of SALM1 in PIP2/F-actin dependent presynaptic assembly.

Reply: The reviewer is right. The RAKA mutant is required to prove causality. We have now performed the proposed rescue experiments expressing SALM1^{RAKA} in SALM1 knockdown neurons. The SALM1^{RAKA} mutant restored total SALM1 intensity in neurons to similar levels as control or SALM1^{wt} rescue conditions (new Supplemental Fig. S2C-D), but did not rescue VGluT1 clustering (new Fig. 7A-C and EV3), synaptic vesicle fusion in the pHluorin assay (new Fig. 8R-W) and evoked EPSC amplitudes (new Fig. 8A-B). These findings support a causal link between the role of SALM1 in synapse development and in promoting F-Actin/PIP2 clustering. We thank the reviewer for this important suggestion.

2. Does SALM1 knock-down in neurons disperse Neurexin1beta at presynaptic, vGluT1-positive sites along neurites and can this be rescued by the SALM1 RAKA mutant (or not)? This would support a requirement of SALM1 for synaptic retention or clustering of Neurexins.

Reply: This is an interesting suggestion. To address this, we co-expressed Nrnx1β-FLAG with SALM1 shRNA and compared VGluT1 intensity and Nrnx1β surface expression to rescue conditions with SALM1^{wt} or SALM1^{RAKA}. We observed that surface Nrnx1β intensity was reduced at VGluT1-positive sites in SALM1 depleted neurons. Rescue with full length SALM1 fully restored surface Nrnx1β intensity at VGluT1-positive sites. In contrast, the SALM1^{RAKA} mutant did not restore Nrnx1β-FLAG intensity (new Figure 6A-B). These findings further support a role for SALM1 in regulating Neurexin surface distribution. We thank the reviewer for this great suggestion.

Minor points.

3. Does the SALM1-induced clustering of Neurexin1beta in turn cluster Neuroligin-1 at cell-cell contact sites? This would be informative to assess roles of SALM1 in trans-synaptic assembly.

Reply: We agree that it would be interesting to further investigate how SALM1 affects trans-synaptic assembly. The reviewer suggests to investigate Neuroligin1 clustering. However, Neuroligin1 self-clusters on the surface of HEK cells (Figure 3A) consistent with previous findings that Neuroligin1 forms homomeric cis-interactions via its extracellular domain (Poulopoulos et al., 2012). Instead, we have now performed additional experiments showing that Nrnx1β clustering by SALM1 enhances Neuroligin1-induced synaptogenesis (new Figure 6C-G). This indicates that SALM1-dependent Nrnx1β clusters indeed participate in trans-synaptic Neurexin-Neuroligin mediated synaptogenesis.

4. The use of autapses precludes assessment of whether SALM1 functions pre- or post-synaptically as it is knocked-down on both sides. Fig. 5A-G is a good example for this issue. The images show dendrites with fewer and less intense vGluT1 puncta upon SALM1 knockdown but it is not clear whether this involves presynaptic loss of SALM1. Do the authors have experiments beyond the co-cultures that allow to address the site of SALM1 action for the effects analyzed here?

Reply: The referee is right that the phenotype observed in autapses is the result of the combined pre- and postsynaptic loss of SALM1. We have added an additional statement in the results/discussion sections (p13, 1269 and p17, 1345) to clarify this point. However, we emphasize that the co-culture data provide clear evidence of the separate functions of pre- and postsynaptic SALM1.

5. It would be quite informative if the authors have results how SALM1 promotes Neurexin1beta clustering at F-actin/PIP2 microdomains.

Reply: We agree that it would be interesting to investigate further how Nrnx1 β is recruited to SALM1/F-actin/PIP2 microdomains. Neurexin interacts with CASK via its PDZ binding domain which provides a scaffold for the simultaneous interaction of protein 4.1N with Neurexin and CASK thereby coupling Neurexin to the F-actin network (Biederer and Sudhof, 2001). To test this, we removed the PDZ domain or the complete intracellular domain of Neurexin-FLAG. Unfortunately, these mutants were not expressed on the surface of HEK cells (see figure 1 at the end of this document). This is consistent with previous findings that the PDZ domain is important for Neurexin membrane expression (Gokce and Sudhof, 2013). We can therefore not conclude if the PDZ domain of Neurexin is important for SALM1 mediated clustering of Neurexin.

6. SALM1 knockdown from 9 div does not reduce synaptic transmission, unlike the knockdown starting at 7 div. This is a rather precisely defined window of requirement; can the authors discuss what developmental maturation steps may cause this loss of SALM1 requirement?

Reply: We agree that the effect of SALM1 knockdown between DIV7 and DIV9 is strikingly different. However, knockdown of SALM1 at DIV2 resulted in a more dramatic loss of VGLUT1 puncta compared to DIV7. This indicates a gradual loss of SALM1 requirement with a final cut off between DIV7 and 9. We have added a discussion on this conclusion in the Discussion of the revised manuscript (p17, l349-356). Several arguments can be considered to explain the observation of loss of SALM1 requirement during the second week in culture. First, as SALM1 depletion during the first week in vitro more drastically reduced synapse number compared to the second week in vitro, these results indicate that SALM1 is required for synapse formation, but may be dispensable for synapse maturation and/or maintenance. Second, during synapse maturation in the second postnatal week, a major switch in synaptic protein expression occurs (Petralia et al., 2005). It is possible that SALM1 expression is also subject to this developmental switch and may be downregulated. In addition, the expression of redundant proteins may be upregulated during the developmental switch. SALM2 expression, for example, is upregulated during later developmental stages and shares similar interaction partners with SALM1 (Ko et al., 2006; Wang et al., 2006). It is thus possible that SALM2 may compensate for loss of SALM1 during later developmental stages.

7. Why do the other SALMs not compensate for knock-down of SALM1, is there information about their expression profile in developing neurons?

Reply: SALM1-5 are synaptic proteins with different interaction partners and different functions. For example, only SALM1-3 contain a PDZ binding domain which binds PSD95 (Morimura et al., 2006). In addition, only SALM3 and 5 form hemisynapses in mixed culture assays through trans-interactions with presynaptic RPTPd adhesion molecules (Mah et al., 2010; Li et al., 2015; Choi et al., 2016). Depletion of SALM1-3 and SALM5 protein levels via knockdown/KO reduces synapse numbers (our data and Ko et al. (2006); Mah et al. (2010); Lie et al. (2016)), but SALM4 depletion results in more synapses (Lie et al., 2016). Both SALM1 and SALM2 bind NMDA receptors, but only SALM2 additionally binds AMPA receptors (Wang et al., 2006). Due to their similarity in binding partners (NMDA receptors and PSD95), we considered redundancy between SALM1 and SALM2 (see new text in the Discussion in response to point 6 of this reviewer). However, Ko et al. showed that SALM2 expression is low during early developmental stages (Ko et al., 2006), indicating that SALM1 loss probably cannot be compensated at this stage. Finally, all five SALMs cluster Neurexin, but it is unknown if SALM2-5 are expressed in presynaptic compartments. Hence, the lack of redundancy is probably the result of differences in protein interaction partners and their spatiotemporal distribution. The background of the reviewer's question is probably that he/she would have liked to see this question being addressed in the manuscript. Therefore, we have added the paragraph above to the revised manuscript (p20, l395-405).

8. The abstract states that SALM1 is preferentially presynaptic. Based on the EM quantification, there is no striking enrichment of SALM1 in pre- vs post-synaptic compartments, though, and if one only looks at synaptic membranes, it is actually twice as abundant postsynaptic. The statement in the abstract should be rephrased and this info about pre/postsynaptic membrane abundance can be stated in the text of the Results.

Reply: The referee argues that SALM1 is twice as abundant on postsynaptic compared to presynaptic membranes and is not convinced of a preferential presynaptic localization of SALM1. However, the EM quantification of the subsynaptic localization of SALM1 in figure 1H shows no striking difference in the amount of SALM1 detected on pre- or postsynaptic membranes and ~60% of SALM1 was detected in the presynapse compared to ~40% in the postsynapse. We therefore think that our statement on SALM1 localization is valid. To strengthen the conclusion, we have

performed additional statistical tests. These show a significant difference in SALM1 abundance between pre- and postsynaptic compartments. We have also improved Figure 2E to increase the visibility of the observed differences.

9. Figure 5B/C/D refers to 'synapse number' as counted parameter, but rather shows vGluT1 puncta abundance. The legend can be updated. Also, how was puncta abundance measured, are these puncta per ROI and how was the ROI defined? Why was not a standard measure obtained, e.g. puncta number per dendrite length?

Reply: We agree and updated the figure and accompanying legend to mention 'vGluT1 puncta' rather than 'synapse number'. The number of vGluT1 puncta were measured using the SynD automated synapse analysis program (Schmitz et al., 2011). We also agree with the referee that the puncta number per neurite length is better measure and plotted this parameter in Fig 7 and used this parameter throughout the manuscript.

10. The IP screen did not identify known CASK binding partners like Neurexins, SynCAMs, or Syndecans. This can be stated to communicate that this approach yielded interesting new partners but does not cover them all, as can be expected.

Reply: We added a statement in the results section of the revised manuscript mentioning the absence of known binding partners (p5, 191-93).

11. The Figure Legends can provide the approaches for the overexpression and knock-down in neurons, i.e. lenti vs calcium phosphate. This helps readers to understand the experiments.

Reply: As suggested by the reviewer, we added the approaches used for the overexpression and knock-down experiments to the Figure Legends.

13. The Methods section refers to multiple different types of neuronal cultures used in this study, including autapses. The info which types of culture were analyzed is not provided in the Results and Figure Legends, making assessment of the data a bit of educated guess work. This needs to be added.

Reply: We agree that this aspect has not been clearly explained. We have added the culture types used for the different experiments to both the Results and Figure Legends.

14. There is a typo in Figure 4B, Phluorin instead of pHluorin.

Reply: This is corrected. Thanks.

Referee #2: This reviewer raises 7 major and 11 minor issues. In total, this reviewer suggests 15 new sets of experiments. Most of these are excellent suggestions. We have now performed all these experiments and the results are all in line with our previous conclusions. We think one set of experiments, to perform rescue experiments with all other SALMs, is really beyond the scope of the paper and beyond what is feasible within the resubmission deadline. For one other set, to perform Ca²⁺-imaging in presynaptic nerve terminals, we think we already have strong arguments to conclude that altered Ca²⁺ dynamics cannot explain our findings.

Major Concerns:

1) It is difficult to easily interpret the results because type of cultures used is not clearly stated in the text, figures or figure legends. Also, the experimental approaches in many of the figures and in the text are lacking and only available in the figure legend (e.g. surface staining vs total). The authors need to ensure that all panels are referred to in the text (e.g. 1D (SMI staining) and 3H).

Reply: We apologize for the missing descriptions of the experimental approaches. We have now added these in the text and to any of the figure legends that did not yet describe this. We have also carefully checked that all figure panels are now referred to in the text.

2) While a significant number of figures are devoted to suggest that SALM1 clusters Nrnx1β on the cell surface via a polybasic motif in PIP2 microdomains that is PDZ-binding motif independent, the actual functional consequence of this clustering appears to be negligible. In neurons, the PDZ-binding motif of SALM1 is necessary for synapse formation and synaptic transmission. If the polybasic mutations and RAKA+ΔPDZ mutations are introduced into these assays (unclustered neurexins), how are synapse formation and synaptic transmission impacted? Thus SALM1 appears to intriguingly have PDZ-dependent and PDZ-independent functions and suggest that neurexin clustering is not the primary functional role of presynaptic SALM1.

Reply: The reviewer is completely right. Our data show that SALM1 has “PDZ-dependent and PDZ-independent functions”. We oppose the view that one of these two aspects is “the primary functional role”. Our main message is that these two aspects are both crucial to bridge intracellular (VGluT1-clustering) and extra-cellular (Neurexin clustering and Neuroligin binding) aspects of synapse development. We have performed additional experiments to further strengthen that clustering of Neurexin by SALM1 is important for synapse development. First, co-expression of Neurexin and SALM1 in neurons efficiently increased the number of synapses formed on Neuroligin expressing HEK cells (new Figure 6C-G). This indicates that the additional Neurexin clusters formed by SALM1 participate in trans-synaptic Neurexin-Neuroligin mediated synaptogenesis. Second, we have performed rescue experiments using the SALM1^{RAKA} mutant. SALM1^{RAKA} did not rescue synapse numbers, VGluT1 intensity and vesicle fusion, while SALM1 Δ PDZ rescued VGluT1 intensity and vesicle fusion, but not synapse formation (new Figure 7 and 8). Together, these findings indicate that SALM1’s polybasic region and the subsequent clustering of Neurexin is an important aspect of SALM1’s function in synapse development.

3) The figures of this paper essentially tell two distinct stories (see point 2) and both parts of the story (PDZ-dependent synapse function and PDZ-independent Nrnx1 β clustering) and experiments/conditions to link the two halves should be included in the manuscript.

Reply: Please also see our response to point 2. We have performed two additional sets of experiments to strengthen this link.

4) Antibody: Validate the antibody by using shRNA2 in WB. Do low molecular weight bands disappear? Are the low MW bands membrane proteins? Is that why the shRNA2 ICC still has significant SALM1 surface staining?

Reply: The requested validation of the SALM1 antibody using shRNA2 is shown in Supplementary Figure 2A. We have added an additional reference to this figure in the text. Blocking protein glycosylation in SALM1-Cherry expressing HEK cells results in the loss of the top band of the WB in Figure 2B. This indicates that the lower bands represent immature forms of SALM1, in line with previous findings (Morimura et al., 2006). The ICC data in supplementary figure 2 depict total endogenous SALM1 levels. Please note that shRNA2 efficiently knocked down (~70-80%) endogenous SALM1 levels, but ~20-30% remains. We therefore expected some staining of SALM1 in shRNA2 infected neurons. This is confirmed by the ICC in figure S2 which shows significantly reduced (but not absent) SALM1 staining upon knockdown with shRNA2.

5) While the authors show that co-transfection of SALM1 without its PDZ-binding motif and Nrnx1 β results in neurexin clustering (Fig. S3L), the authors should use this condition in the artificial synapse formation assay along with proper controls to test if clustering of Nrnx1 β on HE293 cells does anything in this assay. Does the presence of SALM1 (or SALM2-5) somehow enhance the number/size of Homer puncta made onto HEK293 cells?

Reply: We agree that this is an important point. We co-expressed SALM1, SALM1^{RAKA} or SALM1 Δ PDZ with Neurexin in neurons and co-cultured these with Neuroligin expressing HEK cells. We expressed SALM1 and Neurexin in neurons rather than in HEK cells to gain a better insight in the presynaptic function of these proteins. These new experiments show that co-expression of SALM1^{WT} and Neurexin increased the number of VGluT1 puncta formed on Neuroligin expressing HEK cells, but co-expression with SALM1^{RAKA} or SALM1 Δ PDZ did not (new Figure 6A-B). This is consistent with the relative inability of both SALM1 Δ PDZ and SALM1^{RAKA} to increase Neurexin clustering on the surface of neurons compared to SALM1^{WT}. For SALM1 Δ PDZ, this reduced ability to increase Neurexin clustering in neurons is in contrast to our findings that SALM1 Δ PDZ clusters Neurexin in HEK cells. However, Seabold et al. previously showed that the PDZ domain of SALMs is required for normal surface expression in neurons (Seabold et al., 2012). We confirmed that SALM1 Δ PDZ surface expression was indeed reduced in our cultures (new Figure EV2F). In contrast, SALM1 Δ PDZ was strongly expressed on the surface of HEK cells (Supplementary Figure S5H). This difference in SALM1 Δ PDZ surface expression between HEK cells and neurons explains why Neurexin was clustered by SALM1 Δ PDZ in HEK cells, but not in neurons.

6) The functional role of Nrnx1 clustering at PIP2 microdomains is unclear. The authors only test SALM1 Δ PDZ 's ability to rescue the shRNA-mediated evoked EPSC phenotype, however, the RAKA mutant should also be tested because, while the polybasic motif appears to be necessary to

facilitate Nrnx1 β clustering, the potential contribution of this motif to morphology and function was not tested.

Reply: Please also see our reply to point 3, we have now performed additional rescue experiments using the SALM1^{RAKA} mutant. We found that SALM1^{RAKA} did not rescue vGluT1 puncta density and intensity. In addition, we found that SALM1^{RAKA} did not rescue synaptic vesicle fusion (new Figure 7 and 8).

7) The authors' conclusion that the striking reduction in basal synaptic transmission is in part due to impaired vesicular release following SALM1 depletion is questionable. First, in Fig 5H, EM shows no change in docked vesicles following SALM1 shRNA, but a decrease in total vesicle numbers. Single action potential evoked EPSCs are commonly thought to reflect vesicles released from the docked pool (RRP) and, to an extent, vesicles involved in mEPSCs are also provided by the RRP. Second, when normalized to the deacidified condition, the SypHy-PH1 experiment showed no differences. Because the shRNA changes the number of presynaptic vesicles/bouton (Fig. 5), it is misleading to directly compare the absolute F300 between scrambled and shRNA. The mobilization of the recycling and reserve pool by 10Hz stimulation will obviously be dependent on the total number of vesicles/bouton. Third, electrophysiological measurements during 10Hz stimulation revealed no changes in the normalized EPSC amplitude and apparently no changes in decay kinetics, indicating that vesicle release is intact. Fourth, late infection of SALM1 shRNA does not alter synapse numbers and does not alter synaptic transmission. Because vesicular fusion is fine-tuned and is highly dynamic, one would assume that depletion of SALM1 at any developmental timepoint would perturb this system, independent of changes in synapse density. Together, it is likely that these changes in basal synaptic transmission can simply be explained by reduced synapse numbers observed in Fig. 4.

Reply: We agree that EPSC amplitude and mini frequency reflect vesicle release from the RRP. As SALM1 depletion does not affect the number of docked vesicles or the release probability, we have therefore adjusted our previous statement that the reduced vesicle fusion may contribute to reduced EPSC amplitude and mini frequency (p. 15 line 307).

We furthermore feel that it is important to show the absolute F300 and Fmax as this supports our EM and ICC data showing a reduction in total vesicle pool size. To increase clarity for the reader, we have now added an additional statement in the text that the 300AP stimulus given in the SypHy vesicle fusion experiment also recruits vesicles from the recycling and reserve pool (p. 15 line 313).

Minor Concerns:

Figure 1

(A) Figure 1D: What kind of preparation is this? Was SALM1 live surface stained? Were the inset vGluT1 image from the representative neuron or acquired from a different neuron? Was the Homer staining was done in separate neuron preps and, if so, there should be a representative neuron image similar to vGluT1. (B) Timecourse of SALM1 staining (Fig 1D) to match the synapse morphology experiments done later in the paper to confirm that SALM1 is expressed presynaptically. (C) The SMI staining is not convincing that endogenous SALM1 is axonal. There is very little/no overlap between the red and green channels. The meaning of SMI staining is not mentioned in the text. (D) Figure 1G: The Immuno EM does not show SALM1 particles near the active zone (as compared to neurexin/neuroligins. See Burch et al., PLOS ONE. 2017).

Reply: (A) We apologize for this omission. These are hippocampal neurons fixed at DIV16 and stained for SALM1, MAP2, Homer and SMI312. We have now added this to the legend. We have ensured that representative neurons and accompanying insets are depicted for each staining.

(B) We have added an additional time point (DIV9) for SALM1 staining (new Figure S3) and have provided higher resolution images for the DIV16 staining (new Figure 2).

(C) We have added a better description of the SMI staining in the text (p. 6 line 108).

(D) Not all detected SALM1 was localized at the active zone, but the EM data in Figure 2 and Supplementary Figure 1 show several SALM1 molecules detected at the active zone.

Figure 2: In the next figure, the authors point out that SALM1 causes the clustering of Nrnx1 β in HEK293 cells. The authors should quantify artificial synapses assay comparing HEK293s expressing Nrnx1 β alone vs SALM1 and Nrnx1 β . Does Nrnx1 β enhance hemi-synapse formation?

Reply: We now show that co-expression of SALM1 and Neurexin in neurons enhances Neuroigin mediated synaptogenesis compared to Neurexin expression alone (new Figure 6A-B). Please also see our responses to major concerns 2 and 5.

Figure 3: Destabilizing F-actin has a dramatic impact on Nrnx1 β clustering, if F-actin polymerization is enhanced, does one see an opposite effect? Does PIP2 depletion prevent Nrnx1 β clustering by SALM1? 3H is not mentioned in the text and is also found in the supplemental figures. Is CASK expressed in HEK cells? If so, would shRNA KD impair Nrnx1 β 's clustering in the presence of SALM1?

Reply: We enhanced F-actin polymerization, as requested, by treating Neurexin and SALM1 co-expressing cells with Jasplakinolide. This resulted in a saturation of the cell membrane with Neurexin as indicated by a drastic increase in Neurexin intensity compared to DMSO treated cells (new Figure S5A-E). The stabilization of F-actin by Jasplakinolide treatment likely results in stabilization of Neurexin on the cell membrane which matches previous findings that Neurexin is linked to the F-actin network (Biederer and Sudhof, 2001).

We now also show that PIP2 depletion, as requested, by membrane targeted Synaptojanin1 overexpression (as in Milosevic et al. (2005)) results in diffuse Neurexin surface expression in SALM1 and Neurexin co-expressing HEK cells, confirming the role of PIP2 in SALM1 dependent Neurexin clustering (new Figure S5F-G).

Finally, CASK is endogenously expressed in HEK cells (Srivastava et al., 2016). To further investigate how Neurexin1beta is recruited to SALM1/F-actin/PIP2 microdomains, we removed the PDZ domain or the complete intracellular domain of Neurexin-FLAG (rather than shRNA knock down, which in our hands produces off target effects). Expression of these mutants led to a very low surface expression and clustering (please see figure 1 at the end of this document), consistent with previous findings (Gokce and Sudhof, 2013). Hence, Neurexin's PDZ domain is indeed important for its surface expression/clustering.

We have added reference to Fig 3H (new Figure 5E) in the text and apologize for this omission.

Figure S3: Overexpression of SALM1 appears to increase the size of PIP2 microdomains in HEK293 cells. Does SALM1 drive the formation of PIP2 microdomains in a PDZ binding motif independent manner that traps Nrnx1 β ? The SALM1 Δ PDZ experiments in S3L suggest that close proximity of SALM1 with Nrnx1 β is not necessary to mediate Nrnx1 β clustering.

Reply: We have performed additional stainings, as requested, showing that PIP2 microdomains are also present at SALM1 Δ PDZ clusters indicating that PIP2 clustering by SALM1 is independent of the PDZ binding domain (new Figure S5J-K).

How microdomains (PIP2) are defined should be explained around line 168-169 instead of on line 190.

Reply: We agree and have now defined membrane microdomains earlier (p.9 line 170-171).

Figure 4: (A) Does the enrichment of SALM1 and Nrnx1 β at PIP2 microdomains in HEK293 cells also occur in neurons? The authors only stain for total actin.

(B) 4A: It is curious that FLAG-Nrnx1 β alone does not traffic to the surface of neurons. Others have seen robust trafficking (Savas, JN. Neuron 2015) and expression of Nrnx1 β can functionally rescue synaptic transmission (Aoto, J. Nature Neuroscience 2015), arguing that these molecules traffic to the plasma membrane/active zone.

(C) Does the ratio of surface/intracellular Nrnx1 β increase with SALM1 compared to no SALM1 and SALM1 RAKA?

(D) Live surface labeling artificially clusters surface proteins. It is surprising that there are fewer Nrnx1 β puncta in the RAKA condition, indicating that less Nrnx1 β made it to the surface of these neurites. Does overexpression of SALM1RAKA block Nrnx1 β surface trafficking? To determine this, it would be important to measure total surface and/or surface recycling of Nrnx1 β -FLAG levels for SALM and SALM1RAKA overexpression conditions.

Reply: (A) The PH-PLC-Cherry reporter used in this study to detect PIP2 and a (poorly validated) PIP2 antibody are currently the only tools to detect PIP2. We found that in neurons, especially in smaller structures like axons, dendrites and synapses, both methods fail to produce reliable signals that can be interpreted towards PIP2 microdomains, as also concluded before (Micheva et al., 2001). Hence, the lack of proper tools prevents assessment of PIP2 microdomains in neurons and confirmation of the evidence we presented in the original manuscript.

(B) Please note that Neurexin-FLAG is expressed at the surface of neurons, i.e., consistent with previous studies. However, the intensity is much lower compared to SALM-Neurexin co-expressing neurons. For our studies, we exchanged the previously used pFluorin tag of Neurexin-pFluorin, which is efficiently expressed at the membrane (Fu and Huang, 2010), with a FLAG-tag, because GFP (and pFluorin by proxy) forms oligomers under physiological conditions (Jain et al., 2001;

Costantini et al., 2012). We reasoned that the pHluorin tag may oligomerize constructs independent of Neurexin and could mask physiological SALM1-dependent processes. The difference between tags may also explain some difference in membrane expression between FLAG- and pHluorin-tagged Neurexins.

(C) The intensity of Neurexin clusters were increased upon co-expression with SALM1, but not with SALM1^{RAKA} (new figure EV2B-E). As total (surface + intracellular) Neurexin intensity was unchanged for all conditions (new figure EV2G-J), this indicates an increase in the surface/intracellular ratio of Neurexin in SALM1 co-expressing neurons.

(D) Please note that surface Nxn1 β -FLAG levels were similar when Nxn1 β -FLAG was expressed alone or co-expressed with SALM1^{RAKA} (new figure EV2B-E). Only co-expression with SALM1 increased Nxn1 β -FLAG. Hence, SALM1^{RAKA} does not affect Nxn1 β -FLAG surface expression.

Figure S4B: SALM2-5 have a polybasic motif and are surprisingly not clustered but rather diffuse. However, these molecules still have the ability to cluster Nrxn1 β . This is confusing and does not support the hypothesis that F-actin/PIP2 trap SALM oligomers, which then clusters Nrxn1 β . Can the authors explain this observation?

Reply: The SALM2-5 staining in Figure S4B (new figure S6B) is indeed diffuse. However, the SALM1 stainings the reviewer refers to are surface stainings, while the SALM2-5 stainings in Figure S4B are total (surface+intracellular) stainings. SALM1, like SALM2-5, also has a diffuse intracellular staining as observed in Figure EV1A-B. As suggested by the reviewer in major point 1, we stated the specific staining used in each figure and figure legend to improve clarity on the methods used. We apologize for not making this clearer in the original manuscript.

Figure 5: SALM knockdown by shRNA1 is ~50% while shRNA2 is ~80%, yet both display similar loss of synapses. Rescue should be performed for shRNA1 to exclude off-target effects. It is unclear why the endpoint for staining changes depending on the time of lentiviral transduction because synapse density and morphology (e.g. on DIV 9, all synapses are on shafts on DIV14 and 16, most excitatory synapses are located on spines).

The authors should show merged ICC images. Also, does presynaptic localization of CASK change in the SALM1 shRNA or SALM1 shRNA + rSALM1 Δ PDZ conditions? Fig. 5B: rescue experiments should be performed.

Fig. 5B, C D: A description of how the analysis was done is needed as it appears that the total number of synapses in an image were quantified (Synapse numbers are usually normalized to unit length because absolute numbers of synapses are dependent on density, which varies from ROI).

Reply: As suggested by the reviewer, we have performed additional rescue experiments for the DIV2-9 time point (new figure EV3). We agree with the reviewer that synapse number per unit length is a better measure and have altered this now throughout the figure. We feel that by showing the ICC images separately, it will be easier for the reader to observe the differences in the VGluT1 stainings for the different conditions.

We used shRNA2 for all experiments addressing the physiological consequence of SALM1 depletion. In these experiments we used SALM1 rescue constructs to exclude off-targets effects.

We have furthermore performed staining for CASK and VGluT1 in SALM1 depleted neurons and rescue conditions. Unfortunately, CASK staining in neurons was heterogeneous and did not allow proper evaluation of CASK expression in SALM1 depleted neurons (please see figure 2 at the end of this document).

We added a better description of the analysis in Fig. 5 (new figure 6) in the revised manuscript.

Figure 6: The readily releasable pool is responsible for the vesicles used for basal synaptic transmission (single evoked synaptic transmission Fig. 6A and miniature transmission Fig. 6C). Typically, the recycling pool and/or reserve pool participate in transmission following prolonged repeated stimuli. In Fig. 5M, the number of docked vesicles (RRP) is unchanged. The observed change in total vesicle numbers (Fig. 5I) is likely a change in recycling vesicles and/or reserve pool, however, the electrophysiological methods applied in Figure 6A-F do not test the contribution of the recycling pool. It is thus unclear how the change in total vesicle numbers without a change in docked vesicles can contribute to the synaptic phenotype following SALM1 KD.

(A) The significance of SALM1 mediated clustering of beta neurexin-1 has not been tested. Instead, the entire functional phenotype is dependent on the PDZ binding motif of SALM1. Rescue experiments with RAKA mutant for evoked EPSC and phluorin experiments should be performed.

(B) Can SALM2-5 rescue the SALM1 KD phenotype? When overexpressed, are these family members redundant?

(C) It is confusing why infection of shRNA1 at DIV7 significantly reduces synapse numbers at DIV 14, but fails to manifest as a change in mEPSC frequency but does impair evoked EPSCs.

(D) The RRP should be measured.

(E) Are presynaptic calcium dynamics altered by SALM1 KD? The decay tau of the train stimuli should be quantified and the train duration should match the phluorin experiment to test if impaired vesicle fusion is observed electrophysiologically.

(F) Interpretation of Phluorin experiments should be performed on the data normalized to NH₄Cl. It is not accurate to compare absolute ΔF values. The authors should perform additional experiments to test their hypothesis that vesicle fusion is impaired in by SALM1 shRNA because the functional electrophysiological data argue that vesicle fusion is not the primary cause of the phenotype.

Reply: We agree that the functional significance of SALM1-dependent Neurexin clustering was poorly developed in the original manuscript (A) To investigate this, we have now performed new rescue experiments with SALM1^{RAKA}. SALM1^{RAKA} did not rescue the reduced evoked EPSC amplitude and the reduced vesicle fusion upon SALM1 depletion (new Figure 8).

(B) The redundancy among SALM proteins is a complicated issue. Please see our response to minor point 7 of referee #1. Briefly, SALM1-5 are synaptic proteins with different interaction partners and different functions (Ko et al., 2006; Morimura et al., 2006; Wang et al., 2006; Mah et al., 2010; Li et al., 2015; Choi et al., 2016; Lie et al., 2016). Due to their similarity in binding partners (NMDA receptors and PSD95), we considered redundancy between SALM1 and SALM2 (see also our response to point 6 of the referee). However, Ko et al. showed that SALM2 expression is low during early developmental stages (Ko et al., 2006), indicating that loss of SALM1 cannot be compensated by SALM2 during early development. Furthermore, although we show that all five SALMs are able to cluster Neurexin in HEK cells, it is unknown if SALM2-5 are expressed in presynaptic compartments. We therefore feel that the lack of redundancy between the five SALMs may be the result of differences in protein interaction partners and differences in their spatiotemporal distribution. Given the complexity of this issue and the limited availability of tools, we think that a full analysis of the redundancy between all five SALM proteins in neurons is beyond the scope of this study and beyond what can be produced within the resubmission deadline.

(C) Please note that the variability in mini frequency is exceptionally high compared to the EPSC amplitude and synaptic density, especially in the shRNA1 condition. This likely relates to the fact that shRNA1 knock down efficiency is only ~50%. Combined with a low N in the electrophysiology experiments compared to the synaptic density quantifications, this may explain the high variability. No major conclusion was drawn from the mini data and in general the shRNA1 plays a minor role in establishing our main conclusion.

(D) We have now measured the RRP as requested and added these data to the revised manuscript (new Figure 8J). In brief, we observed a smaller pool in SALM1 knockdown neurons versus control neurons.

(E) We have calculated the decay tau of the train stimuli, as requested (new Figure 8G-H). Paired pulse ratio's were unaltered and the normalized rundown kinetics were also highly similar between SALM1 knockdown and control conditions. In addition, the initial release probability calculated from the train stimulus response was unchanged (new Figure 8K). These are strong indications that presynaptic calcium dynamics are not altered in the remaining synapses. We have now added such a statement in the discussion (p. 18 line 360).

(F) As suggested in major concern 7, we have now adjusted our previous statement that the reduced vesicle fusion may contribute to reduced EPSC amplitude and mini frequency (p. 15 line 303). We think it is important to show the absolute F₃₀₀ and F_{max} as this supports our EM and ICC data showing a reduction in total vesicle pool size. To increase clarity for the reader, we have now added an additional statement in the text that the 300AP stimulus given in the SypHy vesicle fusion experiment also recruits vesicles from the recycling and reserve pool.

Referee #3

This reviewer raised 4 major points and 2 minor concerns and states: "I am very enthusiastic about the results and the quality of the demonstration". We thank the reviewer for his/her positive remarks and helpful suggestions.

Major Points:

1- The statistical tests that the authors used for each of their quantification is not specified. There is only a general mention in the method section, which is not sufficient. It is important to know if they can perform the multiple comparisons.

Reply: We apologize that the statistical methods were not specified sufficiently. We have now provided a more detailed description of how the statistics were performed in the Methods section. We have also given a more detailed description of each statistical test used per data set in the figure legends.

2- The mention of the number of cells and the number of culture helpful but it is not clear which N is used for quantification (An average per cell or an average per culture?). In figure 5 for the quantification of the EM, I could not work out what the numbers in the bar refer to. Ideally the average should be per culture to avoid problems associated with artificial increase of the power of the statistical tests.

Reply: We have updated the figures and figure legends to improve clarity on the definition of the average. Briefly, with 'N' we define the number of independent observations, e.g. the number of cultures. While 'n' defines the total number of observations, e.g. the number of neurons (or the number of synapses in case of the EM).

3- I may have missed it but I did not see how many times the western blot in figure 1 (IP) was replicated.

Reply: We have performed 3 co-IP replicates to show the interaction between CASK and SALM1. We have now added this to the figure legend and apologize for the omission.

4- I think the authors should include a negative control for their antibody in Figure 1. They have a siRNA that efficiently decrease the expression of SALM1 using western blot and it would be important to see the effect of the siRNA on the immunostaining.

Reply: The antibody used in Figure 1 is also used to test the efficiency of the shRNAs and rescue constructs on western blot and immunostaining in Supplemental Figure 2. To make this point clearer, we have added an additional reference upon first describing the antibody to Supplemental Figure 2.

• minor concerns:

1- In their experiments in figure2 they authors conclude that knocking down SALM1 interfere with Neurexin and Neuroligin function leading to a decrease in pre or postsynaptic clustering. An alternative explanation is that SALM1 knockdown would decrease Neuroligin or Neurexin expression levels leading to a decrease in clustering. The authors should control for Neurexin and Neuroligin expression levels in SALM1 KD cultures.

Reply: The referee raises an important point that SALM1 knockdown may interfere with Neurexin/Neuroligin functions by affecting Neurexin or Neuroligin expression levels. Unfortunately, available Neurexin/neuroligin antibodies fail to detect neurexins or neuroligins on WB from cultured SALM1-depleted neurons. However, co-expression of SALM1-pHI with Nrnx-FLAG increases Nrnx-FLAG surface expression without an effect on total Nrnx-FLAG levels suggesting that SALM1 only increases Nrnx-FLAG surface expression without increasing total Nrnx-FLAG levels.

2- The authors do not explain why they use a SALM-pHI construct. I presume that the overexpression lead to a massive expression of SALM1 in vesicles interfering with the visualisation of SALM1 at the membrane. The use of pHI would quench SALM1 expressed in the vesicles and allow the visualisation of membrane SALM1 only. If this is the case of if the reason is different the authors should be explicit about it. They should also comment on the level of overexpression and how it could interfere with their results.

Reply: Please note that the pHluorin tag is a pH sensitive variant of GFP that can be well detected by GFP antibodies and can therefore be used similarly to a GFP tag in the ICC experiments described in this paper. We initially designed the SALM-pHI construct for additional live imaging experiments unrelated to the current study, which indeed allows us to observe only surface expressed SALM1-pHI.

References:

- Biederer T, Sudhof TC. 2001. CASK and protein 4.1 support F-actin nucleation on neurexins. *J Biol Chem* 276:47869-47876.
- Choi Y, Nam J, Whitcomb DJ, Song YS, Kim D, Jeon S, Um JW, Lee SG, Woo J, Kwon SK, Li Y, Mah W, Kim HM, Ko J, Cho K, Kim E. 2016. SALM5 trans-synaptically interacts with

- LAR-RPTPs in a splicing-dependent manner to regulate synapse development. *Sci Rep* 6:26676.
- Costantini LM, Fossati M, Francolini M, Snapp EL. 2012. Assessing the tendency of fluorescent proteins to oligomerize under physiologic conditions. *Traffic* 13:643-649.
- Fu Y, Huang ZJ. 2010. Differential dynamics and activity-dependent regulation of alpha- and beta-neurexins at developing GABAergic synapses. *Proc Natl Acad Sci U S A* 107:22699-22704.
- Gokce O, Sudhof TC. 2013. Membrane-tethered monomeric neurexin LNS-domain triggers synapse formation. *J Neurosci* 33:14617-14628.
- Jain RK, Joyce PB, Molinete M, Halban PA, Gorr SU. 2001. Oligomerization of green fluorescent protein in the secretory pathway of endocrine cells. *Biochem J* 360:645-649.
- Ko JW, Kim SH, Chung HS, Kim K, Han KH, Kim H, Jun HJ, Kaang BK, Kim EJ. 2006. SALM synaptic cell adhesion-like molecules regulate the differentiation of excitatory synapses. *Neuron* 50:233-245.
- Li Y, Zhang P, Choi TY, Park SK, Park H, Lee EJ, Lee D, Roh JD, Mah W, Kim R, Kim Y, Kwon H, Bae YC, Choi SY, Craig AM, Kim E. 2015. Splicing-Dependent Trans-synaptic SALM3-LAR-RPTP Interactions Regulate Excitatory Synapse Development and Locomotion. *Cell Rep* 12:1618-1630.
- Lie E, Ko JS, Choi SY, Roh JD, Cho YS, Noh R, Kim D, Li Y, Kang H, Choi TY, Nam J, Mah W, Lee D, Lee SG, Kim HM, Kim H, Choi SY, Um JW, Kang MG, Bae YC, Ko J, Kim E. 2016. SALM4 suppresses excitatory synapse development by cis-inhibiting trans-synaptic SALM3-LAR adhesion. *Nat Commun* 7:12328.
- Mah W, Ko J, Nam J, Han K, Chung WS, Kim E. 2010. Selected SALM (Synaptic Adhesion-Like Molecule) Family Proteins Regulate Synapse Formation. *Journal of Neuroscience* 30:5559-5568.
- Micheva KD, Holz RW, Smith SJ. 2001. Regulation of presynaptic phosphatidylinositol 4,5-bisphosphate by neuronal activity. *J Cell Biol* 154:355-368.
- Milosevic I, Sorensen JB, Lang T, Krauss M, Nagy G, Haucke V, Jahn R, Neher E. 2005. Plasmalemmal phosphatidylinositol-4,5-bisphosphate level regulates the releasable vesicle pool size in chromaffin cells. *J Neurosci* 25:2557-2565.
- Morimura N, Inoue T, Katayama K, Aruga J. 2006. Comparative analysis of structure, expression and PSD95-binding capacity of Lrln, a novel family of neuronal transmembrane proteins. *Gene* 380:72-83.
- Petralia RS, Sans N, Wang YX, Wenthold RJ. 2005. Ontogeny of postsynaptic density proteins at glutamatergic synapses. *Molecular and Cellular Neuroscience* 29:436-452.
- Poulopoulos A, Soykan T, Tuffy LP, Hammer M, Varoqueaux F, Brose N. 2012. Homodimerization and isoform-specific heterodimerization of neuroligins. *Biochem J* 446:321-330.
- Schmitz SK, Hjorth JJ, Joemai RM, Wijntjes R, Eijgenraam S, de Bruijn P, Georgiou C, de Jong AP, van Ooyen A, Verhage M, Cornelisse LN, Toonen RF, Veldkamp WJ. 2011. Automated analysis of neuronal morphology, synapse number and synaptic recruitment. *J Neurosci Methods* 195:185-193.
- Seabold GK, Wang PY, Petralia RS, Chang K, Zhou A, McDermott MI, Wang YX, Milgram SL, Wenthold RJ. 2012. Dileucine and PDZ-binding motifs mediate synaptic adhesion-like molecule 1 (SALM1) trafficking in hippocampal neurons. *J Biol Chem* 287:4470-4484.
- Srivastava S, McMillan R, Willis J, Clark H, Chavan V, Liang C, Zhang H, Hulver M, Mukherjee K. 2016. X-linked intellectual disability gene CASK regulates postnatal brain growth in a non-cell autonomous manner. *Acta Neuropathol Commun* 4:30.
- Wang CY, Chang K, Petralia RS, Wang YX, Seabold GK, Wenthold RJ. 2006. A novel family of adhesion-like molecules that interacts with the NMDA receptor. *Journal of Neuroscience* 26:2174-2183.

Figure 1. Deletion of the PDZ or intracellular domain of Neurexin impairs Neurexin surface expression. Example images of groups of HEK cells calcium phosphate transfected with Neurexin Δ PDZ-FLAG or Neurexin Δ IC. Both constructs were efficiently expressed intracellularly (red), but detection at the surface was very low (green).

Figure 2. Staining for endogenous CASK is heterogenous. Example image of DIV10 sandwich cultured neuron lentivirally infected at DIV3 with scrambled virus and stained for GFP (blue), VGluT1 (red) and endogenous CASK (green). Blue box in merged image indicates area of zoom. CASK staining was largely diffuse with heterogenous intensity across neurites.

Thank you for submitting your revised manuscript to The EMBO Journal. Your study has now been re-reviewed by the three referees and their comments are provided below. As you can see the referees appreciate the introduced changes and support publication here. I also really appreciate the care and the toughness in your response to the referees' comments.

Before I can send you the formal accept letter there are just a few things that should be sorted out.

Regarding referees #3 last point - I also like dot plot presentation and think it is a good way to show data. What we could also do is that you provide the "raw" data in terms of excel files and we can post that data as source data files for the graphs. Happy to discuss further

REFEREE REPORTS:

Referee #1:

The authors have thoroughly addressed my points, both through new experiments and informative responses in their point-by-point replies. I have one comment on a new experiment the authors performed in response to my point 1 regarding causality. The results in Fig.8A/B show that expression of rSALM1 or rSALM1RAKA do not robustly rescue the reduction in EPSC amplitude upon SALM1 knockdown and that the RAKA mutant does not differ from the WT. This could be briefly discussed in one sentence in the Discussion.

Together, this study provides interesting insights into how SALM1 and PIP2 cluster Neurexin molecules and controls their synapse-organizing roles and will be an important contribution to the field.

Referee #2:

The authors have more than adequately addressed my concerns. I thank the authors for such a detailed response and the additional experiments. This is a well done paper with significant importance to the field of cell-adhesion.

Referee #3:

I think the authors did a very good work in addressing the comments of the different reviewer and as a consequence I would recommend this article for publication.

I would like to made a side comment though, which relates to the question of the N and n. The authors draw their conclusions based on a statistical analysis of the n, neurons or synapses, and provide the N, culture as a note. The danger in using the n for statistical analysis is oversampling the dataset. To reassure the reader that this is not the case, I would have recommended to add the average per culture or independaant observation, in a dot plot representation, in the supplementary method. This way the reader can make her/his own mind about the variability between culture.

Referee #1:

The authors have thoroughly addressed my points, both through new experiments and informative responses in their point-by-point replies. I have one comment on a new experiment the authors performed in response to my point 1 regarding causality. The results in Fig.8A/B show that expression of rSALM1 or rSALM1RAKA do not robustly rescue the reduction in EPSC amplitude upon SALM1 knockdown and that the RAKA mutant does not differ from the WT. This could be briefly discussed in one sentence in the Discussion.

Together, this study provides interesting insights into how SALM1 and PIP2 cluster Neurexin molecules and controls their synapse-organizing roles and will be an important contribution to the field.

We thank the referee for the positive comments. We agree with the referee that rSALM1 or rSALM1RAKA did not robustly rescue the reduced EPSC amplitude upon SALM1 depletion. Unfortunately, these specific experiments could not be performed with visual confirmation of lentiviral infection of rescue constructs, which typically is around 80%. Together with the high variability in evoked synaptic responses, this uncertain factor may explain that the average EPSC amplitudes are incompletely restored after SALM1 knockdown and that the RAKA mutant is not significantly different from the WT. As these technical issues only apply to this specific experiment we have added this explanation to the figure legends (p55, 11166-1170) rather than the discussion section to avoid confusion that these issues may apply to other experiments in the manuscript.

Referee #2:

The authors have more than adequately addressed my concerns. I thank the authors for such a detailed response and the additional experiments. This is a well done paper with significant importance to the field of cell-adhesion.

We highly appreciate the positive comments on the revised manuscript.

Referee #3:

I think the authors did a very good work in addressing the comments of the different reviewer and as a consequence I would recommend this article for publication. I would like to made a side comment though, which relates to the question of the N and n. The authors draw their conclusions based on a statistical analysis of the n, neurons or synapses, and provide the N, culture as a note. The danger in using the n for statistical analysis is oversampling the dataset. To reassure the reader that this is not the case, I would have recommended to add the average per culture or independent observation, in a dot plot representation, in the supplementary method. This way the reader can make her/his own mind about the variability between culture.

We thank the referee for the positive comments on the revised manuscript. We have added source data files that contain the raw data of all experiments in the manuscript, so that "the reader can make her/his own mind". See also the editor's first comment.

Comments of the editor:

Regarding referees #3 last point - I also like dot plot presentation and think it is a good way to show data. What we could also do is that you provide the "raw" data in terms of excel files and we can post that data as source data files for the graphs. Happy to discuss further
We have generated reader-friendly source data files for all the data in the manuscript.

3rd Editorial Decision

25th Jun 2019

Thanks you for submitting your revised manuscript to the EMBO Journal.

I have now had a chance to take a look at the revised version and everything looks good.

I am therefore very happy to accept the manuscript for publication here.

Corresponding Author Name:

Journal Submitted to:

Manuscript Number: